# Optimizing 5'UTRs for mRNA-delivered gene editing using deep learning

Sebastian Castillo-Hair[1,2,7], Stephen Fedak[3,7], Ban Wang [4,7],
Johannes Linder [5,6], Kyle Havens[3], Michael Certo[3] & Georg Seelig [1,5] ✉

mRNA therapeutics are revolutionizing the pharmaceutical industry, but methods to optimize the primary sequence for increased expression are still lacking. Here, we design 5'UTRs for efficient mRNA translation using deep learning. We perform polysome profiling of fully or partially randomized 5'UTR libraries in three cell types and find that UTR performance is highly correlated across cell types. We train models on our datasets and use them to guide the design of high-performing 5'UTRs using gradient descent and generative neural networks. We experimentally test designed 5'UTRs with mRNA encoding megaTAL™ gene editing enzymes for two different gene targets and in two different cell lines. We find that the designed 5'UTRs support strong gene editing activity. Editing efficiency is correlated between cell types and gene targets, although the best performing UTR was specific to one cargo and cell type. Our results highlight the potential of model-based sequence design for mRNA therapeutics.

mRNA therapeutics and vaccines provide a safe, effective, and flexible method of delivering transient genetic instructions to living cells and tissues[1]. Compared to plasmid or AAV-based delivery, mRNA offers several advantages, including simple manufacturing that is independent of the encoded therapeutic protein[2], lower immunogenicity, and transient gene expression[3,4]. As a result, mRNA technology has been crucial for the rapid development of vaccines against the COVID-19 pandemic[5,6], and is currently being developed for applications such as protein replacement therapy[7,8], regenerative medicine[9,10], and cancer immunotherapy[11,12], among others[13]. An intriguing use of the mRNA platform is the delivery of gene editing reagents[3,14], because transient expression of gene editors avoids deleterious effects from prolonged exposure, such as off-target editing[4] and reduces the likelihood of forming anti-drug antibodies, thereby allowing for repeated dosing[15]. Though there are multiple gene editing platforms, single-chain compact enzymes such as megaTALs[16] are particularly well-suited to mRNA delivery. megaTALs are fusions of a minimal transcription activator-like (TAL) effector domain with an engineered meganuclease. The TAL effector addresses the meganuclease, which has intrinsic specificity for

a few genomic target sites, to a single site where it catalyzes the formation of a DNA double-stranded break, thereby achieving high activity and specificity[16]. Because of these features, megaTALs have been developed for a number of therapeutically-relevant targets[17–19].

The recent success of mRNA vaccines and therapeutics is the result of decades of research in areas such as lipid nanoparticles for delivery[20], modified nucleosides for decreased immunogenicity[21,22], 5'-cap analogs for improved translation and stability[23], and codon optimization[24]. Comparably little attention has been directed to optimizing untranslated regions (UTRs) despite their roles in controlling mRNA translation and stability. Many mRNA therapies currently utilize UTRs from the alpha- and beta-globin genes or slight modifications thereof, owing to being well described and associated with highly expressed proteins[1,13]. However, increased protein expression has been observed in a few studies when using alternative UTRs[25–28], demonstrating the remaining untapped potential to optimize expression. A major obstacle is the difficulty in predicting the effects of arbitrary UTR sequences, as some cis-regulatory elements can affect multiple molecular processes[29] and interact with RNA-binding proteins[30] and

[1]Department of Electrical & Computer Engineering, University of Washington, Seattle, WA, USA. [2]eScience Institute, University of Washington, WA Seattle, USA. [3]2seventy bio, Cambridge, MA, USA. [4]Department of Biology, Stanford University, Stanford, CA, USA. [5]Paul G. Allen School of Computer Science & Engineering, University of Washington, Seattle, WA, USA. [6]Present address: Calico Life Sciences LLC, South San Francisco, CA, USA. [7]These authors contributed equally: Sebastian Castillo-Hair, Stephen Fedak, Ban Wang. ✉e-mail: gseelig@uw.edu

microRNAs[31,32] that may even be differentially expressed across tissues. Recently, quantitative models based on deep learning that predict translation efficiency[33,34] and mRNA stability[35] from sequence have started to emerge. Using these to guide UTR sequence design for mRNA therapeutics remains an intriguing yet relatively unexplored alternative[36] (Fig. 1A).

The 5'UTR sequence in particular is a major determinant of translation efficiency and thus an intriguing target for engineering[37–39]. To initiate translation, the ribosomal 43 S pre-initiation complex (PIC) scans the 5'UTR in the 5'-to-3' direction until a start codon is found. Therefore, 5'UTRs can affect translation by capturing PICs prematurely via upstream start codons (uAUGs) and ORFs (uORFs)[38,40], interfering with PIC scanning via stable secondary structure[39], or even by directly recruiting ribosomes via Internal Ribosome Entry Sites (IRESs)[41]. Some 5'UTR cis-regulatory elements are exclusively located within a few

bases from the 5' end. For example, 5'-Terminal Oligo Pyrimidine (5'TOP) motifs consisting of a cytosine at position +1 followed by 4 to 15 pyrimidines[42], 5'TOP-like motifs located within four nucleotides of the transcription start site[43], and pyrimidine-rich translational elements (PRTEs) consisting of a uridine flanked by pyrimidines[44,45] upregulate translation in response to mTOR activation during stress and have been linked to cancer initiation and progression. Transcriptome-wide translation measurements in a panel of cell lines[46] and during neuronal differentiation[47] have suggested that 5'UTRs regulate translation in a mostly cell type-independent manner, whereas 3'UTRs have a greater cell type-specific effect. However, some 5'UTRs have been observed to act in a cell type-specific manner, for example, during embryo development[48,49]. To predict the influence of 5'UTR sequence on translation, we previously developed Optimus 5-Prime, a convolutional neural network trained on translation efficiency

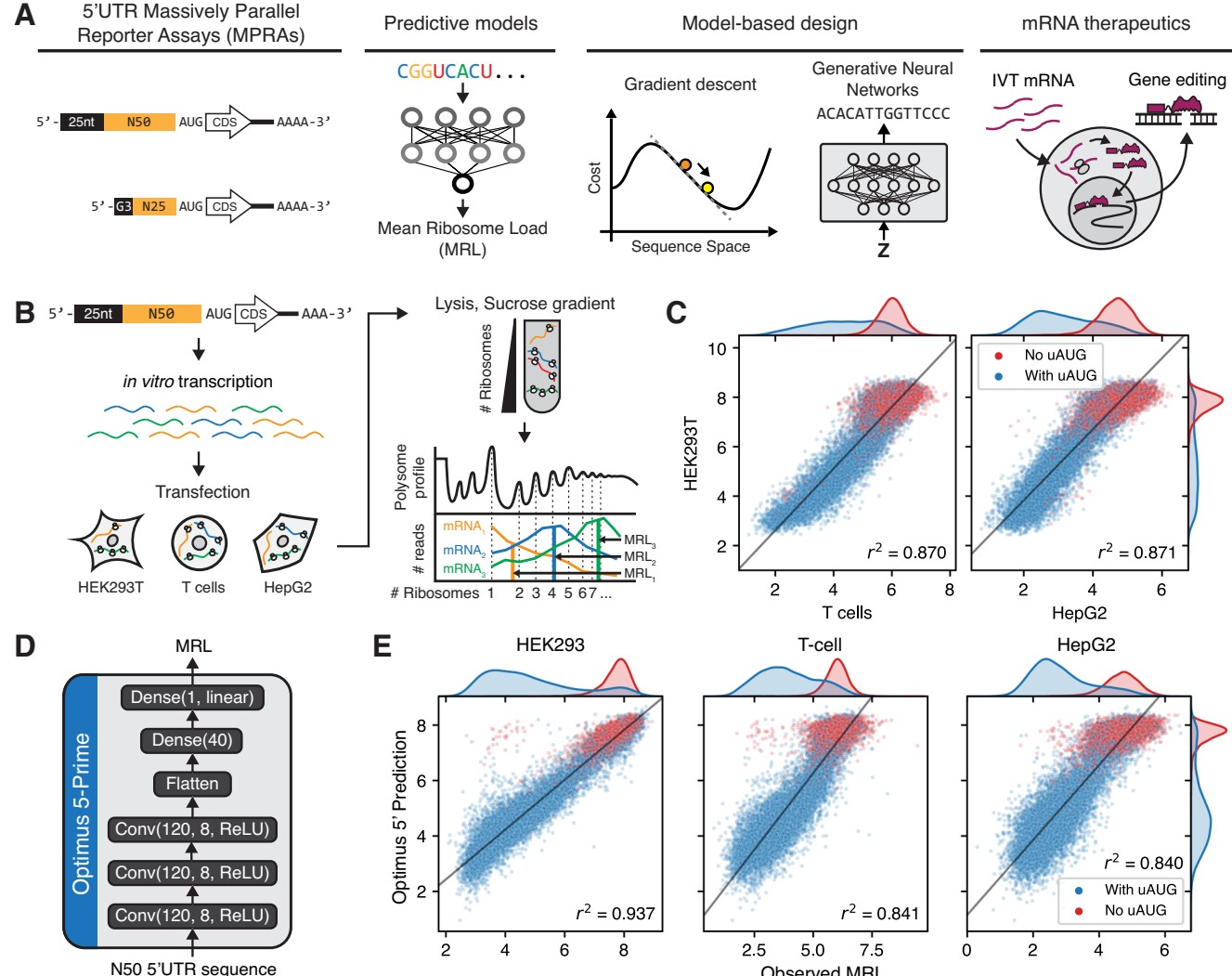

**Fig. 1 | Massively Parallel Reporter Assays (MPRAs) to measure cell type-specific 5'UTR regulation of translation. A** A model-based design strategy for 5'UTRs in mRNA therapeutics applications, using neural network-based predictive models trained on MRPA data. **B** Summary of polysome profiling MPRA. A library with a randomized 50nt 5'UTR region was synthesized as in vitro transcribed (IVT) mRNA, transfected into HEK293T, T cells, and HepG2 cells, and fractionated using a sucrose gradient to separate mRNAs with different numbers of ribosomes. Fractions were then barcoded and sequenced, and the Mean Ribosome Load (MRL) was calculated for each 5'UTR variant as a proxy of translation efficiency. The resulting data contained 204,803 5'UTR variants with 100 or more reads in all replicates, in two replicates in HEK293T, two in T cells, and one in HepG2. **C** Comparison of MRL measurements across cell lines. 5'UTR variants were sorted by the minimum number of reads across all replicates, and the top 20,000 were used for this analysis. Data from only one replicate per cell line is shown. Data from additional replicates can be found in Supplementary Fig. 1. **D** Architecture of Optimus 5-Prime, a convolutional neural network model for predicting MRL from 5'UTR sequence[33]. **E** Optimus 5-Prime predictions compared to MRL measurements in all three cell lines. The top 20,000 5'UTRs by read count in HEK293, which were not used for model training, were used for this analysis. Source data are provided as a Source Data file.

measurements from a synthetic reporter library of 280,000 random 5′UTRs[33]. However, there are limitations to Optimus 5-Prime. First, the reporter design included a constant 25 nt-long region at the very 5′ end of the transcript and Optimus 5-Prime may not have learned to properly model the influence of sequence elements specific to this region[42–45]. Moreover, the overall length of the tested 5′UTRs was 75 nt (25 fixed and 50 random nt) in most of our experimental assays, while for mRNA technology applications, it may be desirable to shorten the 5′UTR to minimize the overall transcript length and to reduce the likelihood of unintentionally including cis-regulatory information that impacts mRNA stability or translation. Second, predictive models used for mRNA therapeutics sequence design should aim to be accurate in all cell types and tissues where the therapy is expected to be functional but it is unclear whether Optimus 5-Prime predictions can generalize beyond HEK293T cells, where our reporter assays were conducted. Third, while we previously used Optimus 5-Prime to guide the design of synthetic 5′UTRs, these sequences were validated through GFP expression and ribosome loading experiments but not in a functional assay relevant to mRNA therapy or related applications[33].

In this study, we designed de novo 5′UTRs for an mRNA-encoded gene editing application using Optimus 5-Prime, thereby validating our previously established deep learning modeling and sequence design methods in the context of a functional assay. We first sought to characterize whether Optimus 5-Prime generalized to two new cellular contexts relevant to mRNA therapeutics. We targeted cultured hepatocellular carcinoma (HepG2) cells as a proxy for liver cells, for which protein replacement[8], regenerative[50], and gene editing[51] mRNA therapies are currently being developed. Additionally, we characterized T cells, where therapeutic mRNA has been used for transient expression of chimeric antigen receptors (CAR)[52] and to express gene editors to knock out specific receptors during manufacturing of allogeneic CAR T cells[53]. Moreover, to study 5′UTR regulation specific to the 5′ terminal region, we constructed and characterized new mRNA reporter libraries with shorter, 25nt or 50nt-long, completely randomized 5′UTRs, and used the resulting data to train a new predictor, Optimus 5-Prime(25). We then designed 5′UTRs for mRNA-delivered megaTAL gene editing therapeutics and conducted gene editing assays in K562 cells to validate them. We used both versions of Optimus 5-Prime along with two design methods we recently developed: Fast SeqProp, based on gradient descent optimization[54], and Deep Exploration Networks (DENs), based on generative neural networks[55]. We found that 24 out of 29 de novo UTRs designed to maximize mean ribosome load (MRL) resulted in high editing efficiency compared to endogenous controls for two different megaTALs. Furthermore, maximum editing activity was achieved with one of the DEN-designed 5′UTRs for one of the megaTAL targets. Finally, we directly measured the MRL and stability of megaTAL-encoding mRNAs for a subset of our engineered 5′UTRs, and found that sequences with high MRL but low gene editing efficiency had a shorter lifetime and a large proportion of ribosome-free (i.e. translationally inactive) molecules. Our results highlight the potential of current sequence design methods for mRNA therapeutics and outline limitations of our current translation predictive models.

## Results

### Optimus 5-prime predictions generalize to cells relevant to mRNA therapeutics

We performed Massively Parallel Reporter Assays (MPRAs) to measure translation efficiency from our previously developed 5′UTR reporter library in HepG2 and T cells (Fig. 1B), following an identical procedure as we did with HEK293T[33] (Methods). Briefly, our library comprised in vitro transcribed (IVT) mRNAs, with a 5′UTR containing an initial constant 25nt-long segment followed by a 50nt-long fully random region, an EGFP CDS, and a 3′UTR derived from the bovine growth hormone (BGH) gene. We transfected the IVT mRNA library and, after an 8 h incubation period, extracted cell lysates in the presence of the

translational inhibitor cycloheximide, performed polysome profiling to separate polysome fractions, and sequenced each fraction. As a proxy for translation efficiency, we calculated the Mean Ribosome Load (MRL) for each 5′UTR, by multiplying the normalized read count on each fraction by the corresponding number of ribosomes[33].

After filtering for sequences with at least 100 reads in all datasets, we obtained translation measurements from 204,803 5′UTR variants in common across five replicates over three cell types, with similar quality and read coverage (Supplementary Fig. 1A, B). Analyzing a subset of the 20,000 sequences with the highest coverage, we found MRLs to be highly correlated across cell lines (Fig. 1C, Supplementary Fig. 1C), with coefficients of determination between cell lines ($r^2 = 0.837$–$0.870$ for HEK293T versus T cells and $r^2 = 0.847$–$0.871$ for HEK293T versus HepG2) comparable to those across T cell replicates ($r^2 = 0.814$, Supplementary Fig. 1C). While these were lower than the HEK293T inter-replicate correlation ($r^2 = 0.938$), this difference could be at least partially explained by the higher HEK293T data quality resulting from a larger number of cells ($> 4$ million vs 1 million) and amount of IVT RNA (14.5 ug vs 1 ug) used in our previous study. Moreover, while $r^2$ decreased as we included more sequences with lower coverage, likely an artifact of decreasing data quality, their relationship across cell lines was maintained (Supplementary Fig. 1D).

Next, we compared these measurements to Optimus 5-Prime predictions (Fig. 1D). While the highest correlation was observed with HEK293T measurements ($r^2 = 0.937$ on 20,000 sequences with the highest read coverage held out from training, Fig. 1E), correlations with measurements in T cells and HepG2 were also high ($r^2 = 0.841$ and $0.840$ respectively). For both HEK293T and T cells, performance was close to inter-replicate correlation, thus absolute differences in $r^2$ are likely due to the higher data quality in HEK293T. Prediction accuracy did not consistently increase when retraining Optimus 5-Prime individually in each cell line (Supplementary Fig. 2) or when training a single multi-output model to predict on all cell lines simultaneously (Supplementary Fig. 3). Finally, given that the most influential known regulatory elements are composed of three letters (AUG, CUG, GUG, etc), we investigated whether any 3-mers could have differential effects over MRL in different cell types. To this end, we trained simple 3-mer-with-position linear models on each replicate (Supplementary Fig. 4A, B) and analyzed the resulting weights, but failed to find any cell line differences beyond those present in replicates of the same cell line (Supplementary Fig. 4C). Together, these results show that observations from our polysome profiling MPRA in HEK293T, as well as predictions from Optimus 5-Prime, generalize to HepG2 and T cells.

### De novo designed 5′UTRs enable high gene editing efficiency from megaTAL-encoding mRNAs

Next, we used Optimus 5-Prime to design de novo 5′UTRs, with the goal of maximizing megaTAL expression from mRNA vectors and therefore improving gene editing efficiency. Specifically, we used megaTALs designed to disrupt two genes whose knockout in engineered T cells enhance antitumor activity[56,57]. The first megaTAL targeted exon 6 of the *TGFBR2* gene, which codes for the TGF-β receptor II, a receptor for the TGF-β cytokine with prominent roles in development, regeneration, immune cell differentiation, and cancer[58,59]. Our second megaTAL targeted exon 1 of the *PDCD1* gene, which codes for the signaling receptor Programmed Cell Death Protein 1 (PD-1) which acts as an inhibitory checkpoint during T cell activation[60].

We designed 19 de novo 5′UTRs and selected 11 control sequences, incorporated them into megaTAL-encoding mRNAs, and quantified their performance via gene editing assays (Fig. 2A, B). Our controls included eight sequences previously measured in our polysome profiling MPRA[33], including four with low or medium measured MRLs as well as four selected from the top 0.02% by measured MRL (Supplementary Fig. 5A). As additional controls, we included the 5′UTR sequences of the human *VAT1* and *LAMA5* genes which were identified

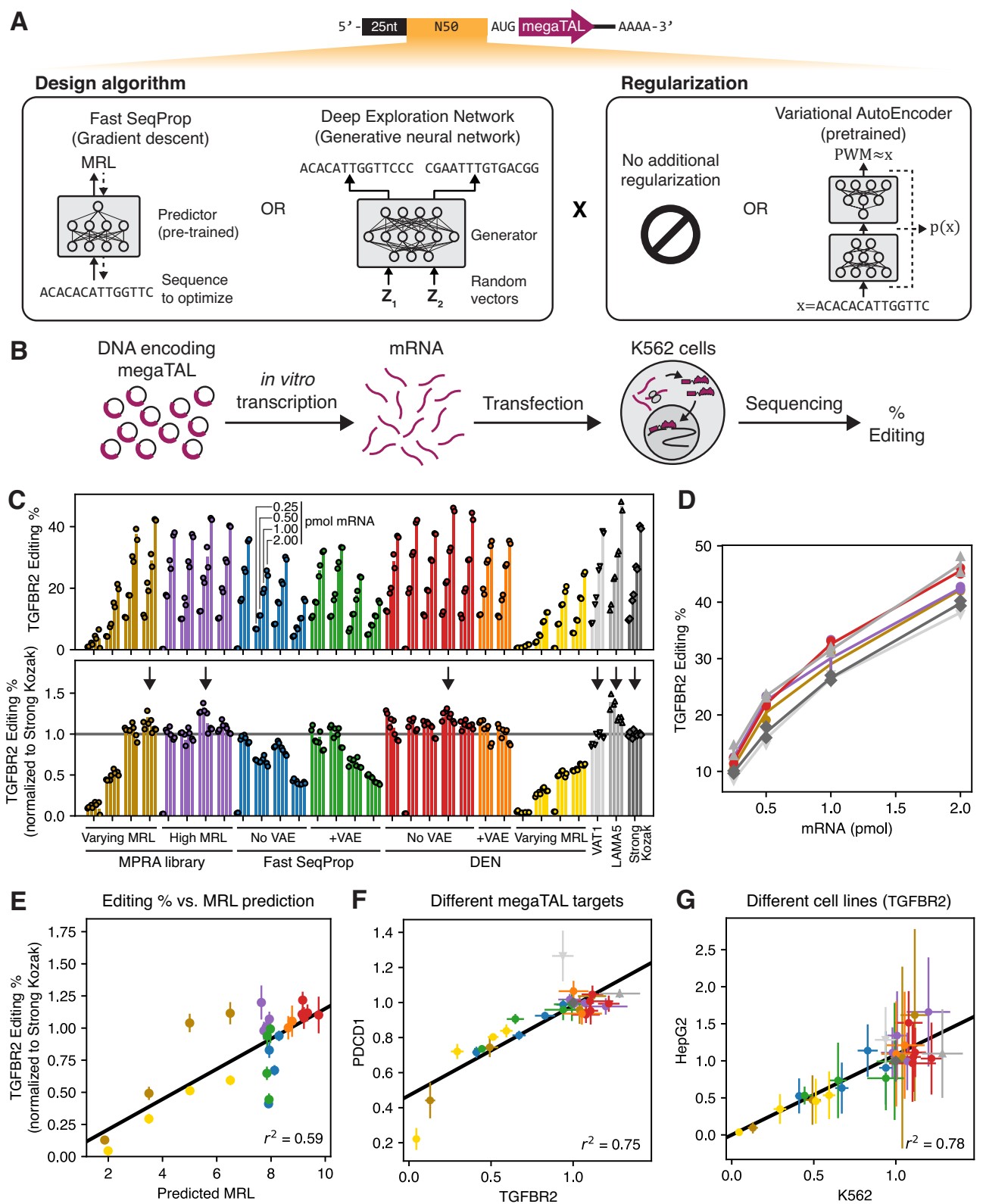

in a prior gene editing screen as the best performing natural 5′UTRs. In previous MPRA measurements[33] these UTRs showed high translation efficiencies similar to the commonly-used beta-globin UTR (top ~20% among ~17,000 short endogenous UTRs, Supplementary Fig. 5B). We also included a minimal 5′UTR consisting of nothing more than a strong Kozak sequence[61], which we had previously found to result in high editing efficiency. All UTRs were preceded only by the initial guanine triplet inserted by IVT.

De novo 5′UTRs were designed using either Fast SeqProp[54] or DENs[55] (Fig. 2A). To preserve Optimus 5-Prime prediction accuracy, 5′UTR architecture was kept identical to the MPRA library used for training: a constant 25nt-long initial region followed by a variable 50nt segment. In Fast SeqProp, a candidate sequence is iteratively refined by following the gradient of the Optimus 5-Prime-predicted MRL with respect to a continuous representation of the sequence. By following the gradient instead of scoring multiple random mutations at each

**Fig. 2 | Model-based design of 5'UTRs for gene-editing mRNA therapeutics.**
**A** Top: schematic of megaTAL mRNA vector. 5'UTRs have the same architecture as our MPRA (Fig. 1B). Bottom: variable 5'UTR regions are designed via a combination of two design algorithms (Fast SeqProp[54] or Deep Exploration Networks (DENs)[55]) and two regularization strategies (no regularization or Variational AutoEncoders (VAEs)[62]). **B** Schematic of gene editing experiment. megaTAL mRNA with each 5'UTR was individually synthesized via IVT and transfected into K562 cells. After 72 h, gene editing efficiency was assessed via sequencing of the target genomic region (**Methods**). **C** Editing efficiencies for mRNAs with a megaTAL targeting the *TGFBR2* gene, for 30 different 5'UTRs, including designs and controls. Each group of four bars represents the editing efficiency of one 5'UTR transfected at four mRNA doses (0.25, 0.5, 1, or 2 pmol). Two biological replicates were performed per 5'UTR and mRNA dosage, and are represented by individual markers. Top: absolute efficiencies. Bottom: efficiencies normalized to the Strong Kozak control at the corresponding mRNA dosage. Editing efficiencies for the first High MRL MPRA control, the first No VAE Fast SeqProp design, the second No VAE DEN design, and the third Varying MRL DEN design were close to zero only at 0.25 pmol mRNA, and were deemed to be caused by experimental error and excluded from subsequent analysis. **D** Absolute editing efficiency as a function of mRNA dosage for 5'UTRs indicated with a vertical arrow in the bottom panel of **C**. **E** Kozak-normalized editing efficiency of the *TGFBR2* megaTAL vs. Optimus 5-Prime predictions. **F** Comparison of Kozak-normalized efficiencies with megaTALs targeting the *TGFBR2* or the *PDCD1* genes. **G**) Kozak-normalized efficiencies for the *TGFBR2* megaTAL in K562 versus HepG2. In (**E**), (**F**), and (**G**), markers and error bars represent the mean and standard deviation of Kozak-normalized efficiencies ((**C**), Supplementary Fig. 11A, and Supplementary Fig. 12A, C) across all mRNA dosages for each 5'UTR ($n = 8$ for most 5'UTRs: 2 replicates and 4 dosages. The 0.25 pmol dosage was excluded for four sequences as described above, therefore $n = 6$ for these). Source data are provided as a Source Data file.

step, Fast SeqProp can design high-performing sequences hundreds of times faster than simulated annealing or genetic algorithms, though it may still get stuck in local optima or overfit the predictor[54]. Although out-of-frame uAUGs are unlikely to occur in high fitness sequences, in-frame uAUGs could result in high predicted MRL but would produce an incorrect N-terminus. Therefore, we included a penalty against generated uAUGs (Methods). To reduce the possibility of overfitting, we scored designed sequences using an independently trained linear k-mer model (Methods, Supplementary Fig. 6). By following this procedure, we generated ten candidate sequences, from which we randomly selected four to be tested in our gene editing assays.

To design sequences with DENs, we trained a generative neural network with the objective of maximizing Optimus 5-Prime-predicted MRL while minimizing the similarity across generated sequences (Methods, Supplementary Fig. 7A, B). By explicitly minimizing similarity, we force the generator to capture a large section of the sequence space, thereby reducing the possibility of overfitting or getting stuck in local optima[55]. We generated 1,024 5'UTRs and selected 5 from the top 20 by predicted MRL for gene editing experiments (Supplementary Fig. 7C–E). In addition, to validate the accuracy of the design method and predictor, we trained a DEN in inverse regression mode, where the generator receives an additional input that specifies a target MRL (Supplementary Fig. 8A, B), and designed four 5'UTRs with predicted MRLs of 2, 3.5, 5, and 6.5 for experiments (Supplementary Fig. 8C–F).

As design algorithms seek to maximize performance, they may drift into low-confidence sequence space regions of the predictor, where sequences are too dissimilar from the training data and predictions are less accurate. To prevent this, we trained a Variational Auto Encoder (VAE)[62], a neural network that learns the marginal distribution of the training data and estimates the likelihood of any new sequence with respect to this distribution. We then used the estimated likelihood as a regularization penalty to the cost function in both Fast SeqProp and DEN (Fig. 2A, Supplementary Fig. 9A–F, Methods). Specifically, we trained a VAE on a subset of 5,000 5'UTRs selected from our MPRA dataset for their high-measured MRL and read depth (Supplementary Fig. 9G). We then designed ten additional sequences via Fast SeqProp with VAE regularization and selected four for gene editing assays. Finally, we trained a new DEN with VAE regularization, generated 1,024 sequences with high predicted MRL, and picked two from the top 10 for gene editing assays (Supplementary Fig. 10). In summary, 19 de novo 5'UTRs were selected for gene editing assays, including 15 sequences designed for maximal MRL (4 with Fast SeqProp, 4 with Fast SeqProp + VAE, 5 with DEN, 2 with DEN + VAE), as well as 4 UTRs with low and medium target MRLs designed with a DEN. The sequences of all 5'UTRs tested in gene editing assays can be found in Supplementary Data 1.

To evaluate the performance of our designs, we synthesized IVT mRNA containing candidate 5'UTRs followed by the megaTAL CDS, transfected these at four dosage levels (0.25, 0.5, 1, and 2 pmol IVT mRNA) into K562 (lymphoblast) cells, and quantified the percentage of successful non-homologous end joining (NHEJ)-mediated gene disruption via sequencing (Fig. 2B, Methods). As expected, editing efficiency increased with mRNA dosage for all 5'UTRs, with several designs exceeding 40% for the *TGFBR2* and 80% for the *PDCD1* megaTALs at 2 pmol mRNA (Fig. 2C top, Supplementary Fig. 11A top, Fig. 2D). Editing efficiencies normalized against the Strong Kozak control (hereafter Kozak-normalized efficiencies) were highly consistent across all dosage levels (Fig. 2C bottom, Supplementary Fig. 11A bottom). Most of the assayed sequences showed editing efficiencies comparable to the Strong Kozak control. However, 50% of the Fast SeqProp-generated sequences showed lower editing efficiencies despite having high predicted MRL, regardless of VAE regularization. Kozak-normalized editing efficiencies were highly correlated with predicted MRL over all designed 5'UTRs (Fig. 2E, Supplementary Fig. 11B), although Fast SeqProp-derived sequences with low editing efficiency deviated from the linear trend the most. While we found these observations to hold for both *TGFBR2* and *PDCD1* megaTALs (Fig. 2F), the specific 5'UTRs resulting in maximal editing differed: *LAMA5* performed the best and *VAT1* performed similarly to the Strong Kozak control with the *TGFBR2* megaTAL, whereas the opposite was true for *PDCD1* (Fig. 2F). Finally, we repeated our assay in HepG2 cells and, while the general trends in Kozak-normalized efficiency were maintained (Fig. 2G), absolute efficiency was lower and measurement variability was higher (Supplementary Fig. 12).

## Measuring translation efficiency of short, fully variable 5'UTRs

5'UTR regulation may differ when sequence elements are placed close to the 5' terminus. For example, various pyrimidine-rich motifs have been found to influence translation in response to stress when located within a few bases of the 5' end[42–45]. Our previous 5'UTR MPRA was unable to interrogate this region, as a fixed 25nt segment was placed at the 5' end to facilitate library preparation (Fig. 1B). To overcome this limitation, and to enable design of shorter 5'UTRs, we performed polysome profiling MPRAs on two new "random-end" mRNA libraries, where the 5'UTR consisted only of a variable 25nt or 50nt region preceded only by the guanine triplet introduced by IVT (Fig. 3A). As with our previous 50nt "fixed-end" library (Fig. 1B), we transfected these random-end mRNA libraries into HEK293T cells and collected lysates 12 h later. To compensate for a lack of a constant 5' end for PCR-based incorporation of sequencing adapters, we used template switching (TS), wherein a reverse transcriptase derived from the Moloney murine leukemia virus appends three non-templated deoxycytosines after reaching the 5' end of the template mRNA. Then, a template switching oligo with three riboguanines (rGrGrG) in its 3' end binds to the non-templated overhang, thereby becoming the new reverse transcription (RT) template and providing a fixed cDNA sequence for subsequent adapter incorporation (Fig. 3A,

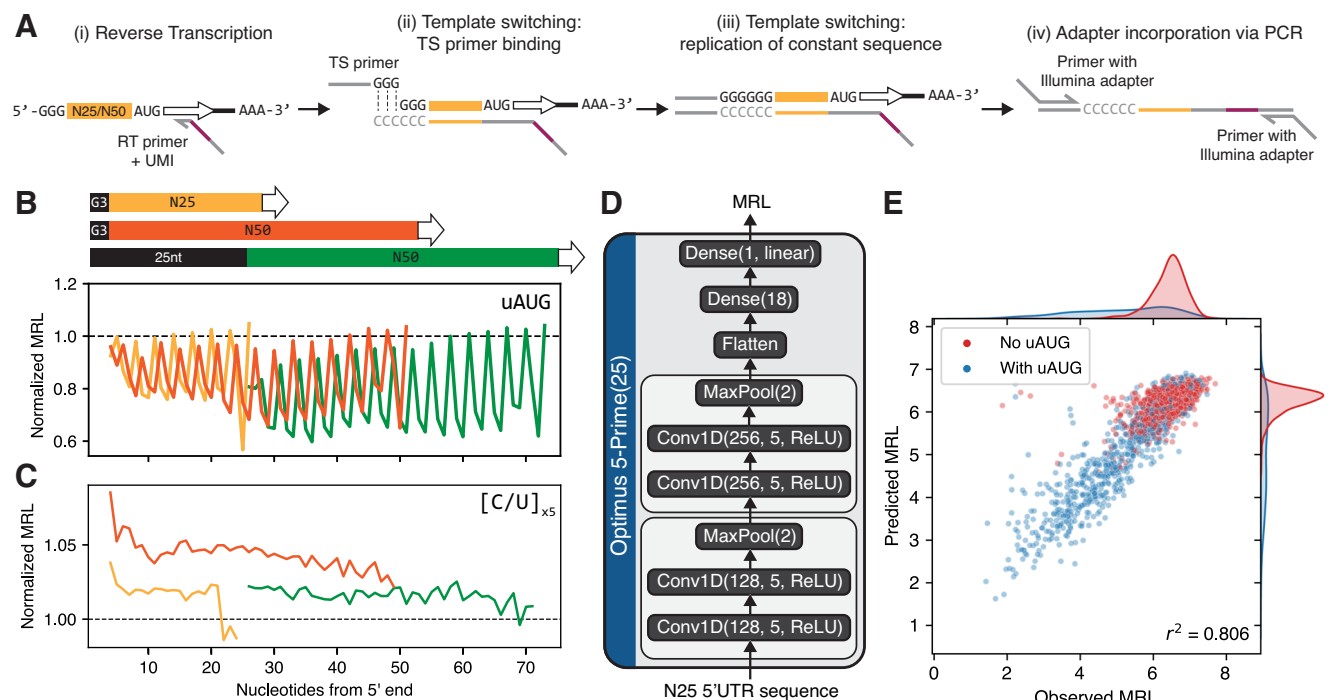

**Fig. 3 | Polysome profiling MPRA on fully randomized 5′UTR libraries.** New libraries contain a 25nt- or 50nt-long randomized region in the 5′UTR, preceded only by the G triplet appended by IVT. **A** Schematic of mRNA library preparation based on template switching (TS). (i) Reverse transcription (RT) proceeds from the EGFP CDS into the mRNA 5′end. (ii) The reverse transcriptase (RTase) adds three deoxycytosines (CCC) to the 3′ of the cDNA, to which a TS primer ending in three riboguanines (rGrGrG) binds. (iii) the RTase switches templates, thereby adding the reverse complement of the TS primer to the 3′ end of the cDNA. (iv) Illumina adapters are incorporated using PCR via primers that bind to the flanking constant regions. **B**-**C** Median MRL of all sequences containing a uAUG (**B**) or a 5nt-long oligopyrimidine (C or U) tract (**C**) at the indicated position from the start of the transcript, for the 25nt- (yellow) or 50nt-long (orange) randomized 5′UTR libraries, as well as our previous fixed-end 50nt library (green). MRL was normalized to the median of each library. **D** Architecture of Optimus 5-Prime(25), trained on data from the random-end 25nt MPRA library. **E** Performance of Optimus 5-Prime(25) on a set of 2,000 5′UTRs from the random-end 25nt library held out from training. Source data are provided as a Source Data file.

Methods). We then performed Illumina sequencing and data processing (Methods), and calculated MRLs from read counts as previously[33].

We performed two biological replicates with the 25nt random-end library and obtained good-quality (sum of reads across replicates greater than 100) MRL measurements from 168,000 distinct UTRs (Supplementary Fig. 13A, B). Inter-replicate MRL correlation was good ($r^2 = 0.692$ for the top 20,000 sequences by read coverage, Supplementary Fig. 13C, D), although lower than our previous fixed-end 50nt library ($r^2 = 0.938$, Supplementary Fig. 1D). Similarly, we performed one replicate with the 50nt random-end library, and obtained MRL measurements from 149,000 sequences at the same quality level (Supplementary Fig. 14). As with our previous fixed-end library, we found that 5′UTRs with uAUGs out of frame with respect to the intended AUG had significantly lower MRL compared to the median of the library and with sequences with in-frame uAUGs (Supplementary Fig. 15A). A similar effect, although of much lower magnitude, was observed for upstream non-canonical start codons (Supplementary Fig. 15B, C). Interestingly, MRL attenuation was noticeably lower when the uAUG was located near the 5′ end in the random-end libraries, suggesting distinct regulation at the very 5′ end compared to the rest of the 5′UTR that could not be captured in our previous fixed-end library (Supplementary Fig. 15A, Fig. 3B).

Next, we evaluated the effect of short pyrimidine tracts (5 x C/U) at different locations within the 5′UTR on measured MRL, using data from both random-end and fixed-end library data (Fig. 3C, Supplementary Fig. 16). We found that pyrimidine tracts generally led to a small but statistically significant MRL increase compared to the library median (Supplementary Fig. 16). For both libraries, we observed a noticeable decrease in effect size with increasing distance of the pyrimidine tract from the 5′ end (Fig. 3C, Supplementary Fig. 16). Therefore, our data is consistent with oligopyrimidine tracts at the start of the transcript resulting in slightly increased translation in HEK293 cells even in the absence of stressors.

## Predicting translation efficiency from short, fully variable 5′UTRs

We next sought to obtain a model that generates accurate predictions on 25nt-long 5′UTRs. We evaluated candidate models via their prediction accuracy on the top 2,000 sequences by read coverage from the random-end 25nt MPRA library, which showed good inter-replicate correlation ($r^2 = 0.844$, Supplementary Fig. 13D). We first tested Optimus 5-Prime, for which 25nt-long input sequences were one-hot encoded and zero-padded on the left to reach the required input length. However, we found its accuracy to be relatively low ($r^2 = 0.564$ for 50nt Optimus 5-Prime, Supplementary Fig. 17A, $r^2 = 0.600$ for 25–100nt Optimus 5-Prime, Supplementary Fig. 17B). We hypothesized that these models, trained on data from 5′UTRs with a constant 5′ region, could not properly account for differential regulatory effects that sequence elements can have when located near the start of the transcript (Fig. 3B, C). Thus, we developed Optimus 5-Prime(25), a model trained directly on the random-end 25nt MRPA data. Inspired by the convolutional network VGG-16[63], Optimus 5-Prime(25) contains two blocks with two convolutional layers and one pooling layer each, followed by two fully connected dense layers that ultimately compute the predicted MRL (Fig. 3D, Methods). This model showed good performance on the same test set which was held out from training ($r^2 = 0.806$ on the top 2000 sequences by read coverage, Fig. 3E).

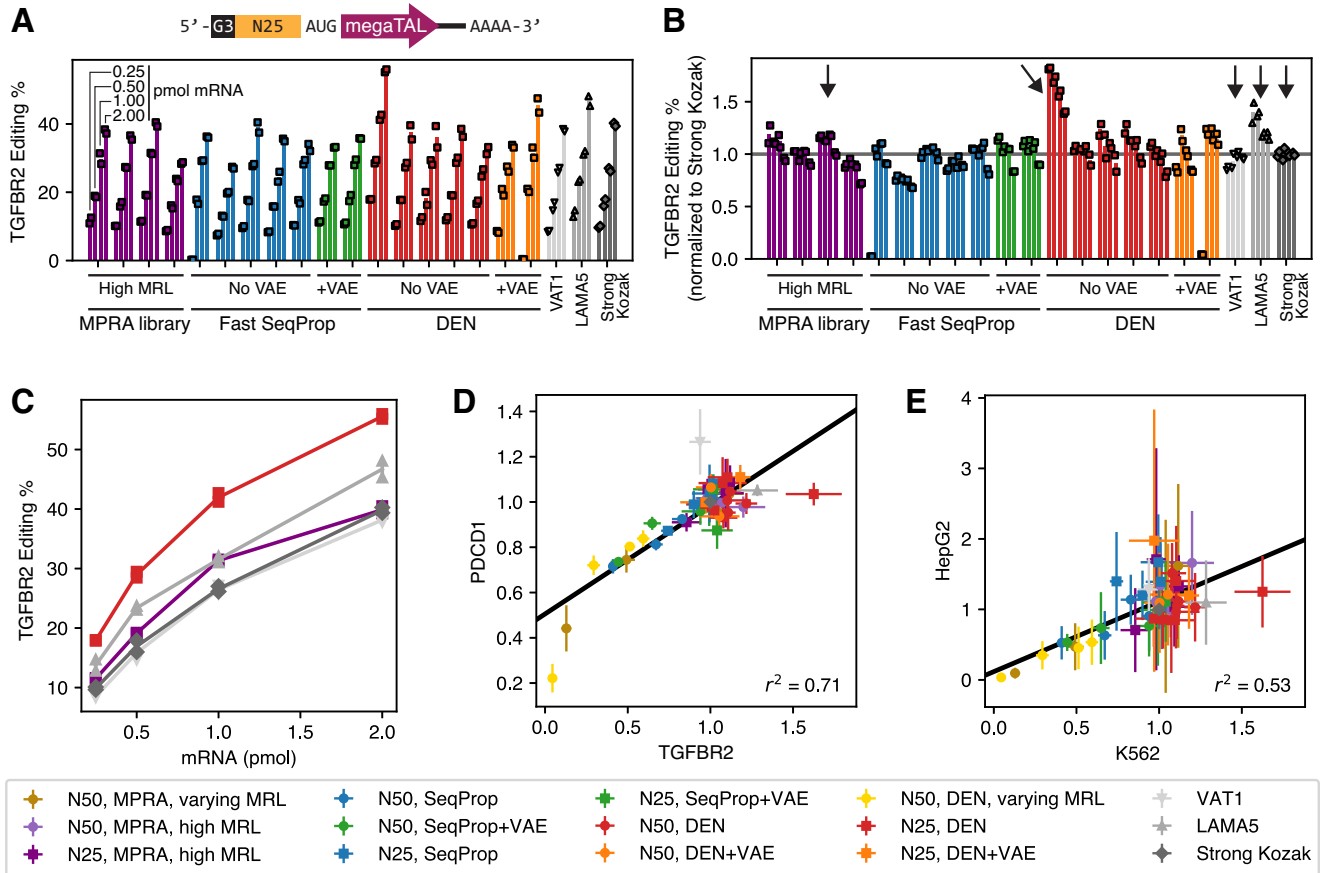

**Fig. 4 | Model-based design of shorter 5′UTRs for gene editing mRNA therapeutics.** (**A**) Top: schematic of mRNA vector, with a 25nt-long variable 5′UTR segment as in Fig. 3. Bottom: Absolute editing efficiencies for mRNAs with a megaTAL targeting the *TGFBR2* gene, for 21 different 5′UTR including designs and controls. Each group of four bars represents one 5′UTR sequence transfected at four mRNA doses (0.25, 0.5, 1, or 2 pmol mRNA). Two biological replicates were performed per 5′UTR and mRNA dosage, and are represented by individual markers. Colors represent the source in the case of controls or the design method. Editing efficiencies for the first No VAE Fast SeqProp design and the second +VAE DEN design were close to zero only at a dosage of 0.25 pmol mRNA, and were deemed to be the result of experimental error and excluded from subsequent analysis. (**B**) Editing efficiencies normalized to the Strong Kozak control at the corresponding mRNA dosage. (**C**) Absolute editing efficiency as a function of mRNA dosage for a few selected 5′UTRs indicated with a vertical arrow in (**B**). (**D**) Comparison of Kozak-normalized editing efficiencies when using a megaTAL targeting the *TGFBR2* gene vs. the *PDCD1* gene. (**E**) Comparison of Kozak-normalized editing efficiencies for the *TGFBR2* megaTAL in K562 cells or HepG2. In (**D**) and (**E**), each marker and error bar represent the mean and standard deviation of the Kozak-normalized editing efficiencies ((**B**) and the bottom panels of Supplementary Fig. 23 and Supplementary Fig. 24) across all mRNA dosages for a particular 5′UTR (*n* = 8 for most 5′UTRs: 2 biological replicates and 4 dosages per replicate. The 0.25 pmol dosage was excluded for two sequences as described above, therefore *n* = 6 for these 5′UTRs only). Data from designs with the fixed-end 50nt architecture (Fig. 2 and relevant Supplementary Figs.) are also included in (**D**) and (**E**). Source data are provided as a Source Data file.

## Designed short 5′UTRs enable high megaTAL-induced gene editing activity

Finally, we used Optimus 5-Prime(25) to design 14 shorter 5′UTRs for our megaTAL mRNAs. As before, we used Fast SeqProp to design ten new 5′UTRs with maximal predicted MRL, validated these against an independent k-mer linear model (Supplementary Fig. 18, Methods), and randomly selected four for gene editing assays. We then trained a DEN to generate 1024 25nt-long 5′UTRs that maximize both sequence diversity and predicted MRL, and selected 5 from the top 25 by MRL (Supplementary Fig. 19, Methods). To test the effect of VAE regularization, we trained a new VAE on 5000 high-coverage, high MRL sequences from the 25nt random-end library (Supplementary Fig. 20, Methods). We then used VAE estimated likelihood as a regularization term to design ten additional 5′UTRs using Fast SeqProp (Supplementary Fig. 18), and randomly selected two for gene editing assays. Similarly, we trained a new DEN with VAE regularization to generate 1024 5′UTRs with maximal predicted MRL, from which we selected two from the top 10 (Supplementary Fig. 21). Notably, despite optimizing based on MRL only, our designed sequences predominantly started

with an oligopyrimidine repeat: from our final selection, 5 out of 7 Fast SeqProp and 4 out of 7 DEN-generated sequences contain TTY after the initial GGG required by IVT, indicating that our methods captured and exploited the enhancing effect of this sequence motif (Fig. 3C). As controls specific to this shorter 5′UTR design, we included four 5′UTRs from the random-end 25nt library with MRLs within the top 0.25% of the library (Supplementary Fig. 22).

We then evaluated the gene editing performance of these designs in K562 as above (Fig. 2B). In practice, we assayed defined-end (Fig. 2C–G) and random-end (Fig. 4) sequences simultaneously to enable direct comparisons across 5′UTR architectures (Methods). As with our previous designs, editing efficiency increased with mRNA dosage (Fig. 4A, Supplementary Fig. 23 top). When normalizing against the Strong Kozak control, we found that most of our designed 5′UTRs performed comparably to the high-performing controls at all mRNA dosages, with the exception of one of the Fast SeqProp designs (Fig. 4B, Supplementary Fig. 23 bottom). Moreover, when targeting the *TGFBR2* gene, one DEN-designed sequence outperformed all other UTRs, including defined-end designs, at all mRNA dosages (absolute

efficiency of 55.6% at 2 pmol mRNA), improving over the previous best performer *LAMA5* by 18–33% (Fig. 4C). When considering both defined-end and random-end designs, Kozak-normalized editing efficiency was highly correlated across the *TGFBR2* and *PDCD1* megaTALs (Fig. 4D). However, the DEN-designed sequence that showed the highest efficiency with the *TGFBR2* megaTAL performed only as well as the Strong Kozak control when combined with the *PDCD1* megaTAL (absolute efficiency of 80.8% at 2 pmol mRNA), with the *VAT1* control performing best in this context (absolute efficiency of 91.4% at 2 pmol mRNA, Fig. 4D, Supplementary Fig. 23). Finally, we repeated these experiments in HepG2 cells and found high Kozak-normalized efficiencies for all new designs, although high replication noise prevented us from identifying a single best performing sequence (Fig. 4E, Supplementary Fig. 24). In conclusion, by using model-based design methods with Optimus 5-Prime(25), we successfully generated de novo 5′UTRs that supported high gene editing activity, including one that outperformed all others for the *TGFBR2* megaTAL.

### Differences in the ribosome-free mRNA fraction and in decay kinetics explain performance discrepancies in maximal MRL 5′UTRs designs

While most 5′UTRs designed to maximize MRL resulted in high gene editing activity, a few had worse performance than expected from their predicted MRL. This discrepancy was particularly pronounced for some of the fixed-end Fast SeqProp designs (Fig. 2C–G). To explain these inconsistencies, we performed polysome profiling on all megaTAL-encoding mRNAs with fixed-end 50nt 5′UTRs (Fig. 5A, Methods), hypothesizing that the actual MRL for FastSeqProp designs might be lower than predicted. Overall, measured MRLs had a higher baseline and a reduced range compared to those from our MPRA experiments and Optimus 5-Prime, possibly due to the longer CDS (2.7 kb vs. 733 bp in EGFP) capable of holding more translating ribosomes and the lower sensitivity of quantifying heavier polysome fractions. Nevertheless, when including submaximal MRL designs, measured MRLs were well correlated with Optimus 5-Prime predictions ($r^2 = 0.53$, Supplementary Fig. 25A) and with editing efficiencies for both the *TGFBR2* ($r^2 = 0.64$) and *PDCD1* ($r^2 = 0.80$) megaTALs. Within sequences designed to maximize MRL, correlation with editing efficiency was lower ($r^2 = 0.40$ for *TGFBR2*, Fig. 5B; $r^2 = 0.35$ for *PDCD1*, Supplementary Fig. 26A), with underperforming FastSeqProp designs having slightly but not dramatically lower measured MRLs. Overall, measured MRLs were not more predictive of editing efficiency than Optimus 5-Prime predictions, and could not adequately explain the lower editing of FastSeqProp designs.

Next, we quantified the proportion of mRNAs in the ribosome-free fraction of the polysome profile. (Fig. 5C). Our MRL definition only considers mRNAs with one or more ribosomes, and a high ratio of ribosome-free to total mRNA abundance could explain low protein output despite a high MRL. For sub-maximal 5′UTR designs, we observed a strong negative correlation between the free-to-total ratio and the measured (Fig. 5D) or predicted (Supplementary Fig. 25B) MRL. This observation is consistent with a model wherein inefficient translation initiation results in both high levels of free mRNA and overall reduced translation. Within maximal MRL designs, the free-to-total ratio was also negatively correlated with editing efficiency (*TGFBR2*: $r^2 = 0.65$, Fig. 5E, *PDCD1*: $r^2 = 0.51$, Supplementary Fig. 26B), although following a different trend line than the submaximal designs. Notably, Fast SeqProp designs with low editing efficiency had the highest free-to-total abundance ratio (Fig. 5E), even above submaximal MRL designs, and even though they were associated with fairly high MRLs. The lower editing efficiency of FastSeqProp designs is thus at least partially explained by the accumulation of the corresponding mRNA in the free mRNA fraction.

Additionally, we characterized mRNA stability by quantifying cellular mRNA abundance at multiple timepoints after transfection (Fig. 5F, Methods). Protein output is proportional to both mRNA abundance and translation, thus variation in mRNA stability could explain variation in editing efficiency. Surprisingly, decay kinetics showed a biphasic behavior which could not be accurately fit to a single exponential decay curve (Supplementary Fig. 27). As an alternative metric for stability, we calculated the integrated mRNA abundance over time (Fig. 5G), reasoning that it should be proportional to the overall synthesized megaTAL protein under a simple protein production kinetic model (Methods). Integrated abundance was generally higher for submaximal MRL 5′UTRs, and, within maximal MRL designs it was well correlated with editing efficiency (*TGFBR2*: $r^2 = 0.55$, Fig. 5H; *PDCD1*: $r^2 = 0.73$, Supplementary Fig. 26C), with low performing FastSeqProp designs showing the lowest abundances. Therefore, the comparably lower stability of FastSeqProp designs further explains their lower editing efficiency.

Finally, we attempted to identify regulatory sequence elements that may drive these behaviors via motif enrichment analysis. Within the six 5′UTRs with the highest free-to-total ratio, we found significant enrichment of motifs CCGUA and UWAGUAG (E value = 2.9e-2) compared to all other sequences assayed here (Methods). Similarly, sequences with the lowest integrated abundance were significantly enriched on motif HCCGUA (E value = 2.9e-2). Further work with a larger number of sequences will be needed to confirm whether these are true drivers of stability and accumulation in the ribosome-free fraction.

## Discussion

In this study, we first obtained MRL measurements from approximately 200,000 5′UTRs across T cells and HepG2 cells; each 5′UTR contained a 25nt constant segment followed by a 50nt fully random region. We found that measurements were highly correlated between the two cell types and with measurements previously performed in HEK293T cells (Fig. 1C, Supplementary Fig. 1C). Accordingly, all measurements were accurately predicted by Optimus 5-Prime, a model trained on HEK293T cell data only. Retraining Optimus 5-Prime on data from each cell line did not consistently improve performance (Supplementary Fig. 2). Therefore, our results suggest that synthetic 5′UTRs regulate translation from IVT mRNA with little specificity across human cells relevant for mRNA therapeutics. This is further supported by our gene editing results: sequences designed using a model trained on HEK293T data resulted in MRLs in K562 cells that were highly correlated with predictions (Supplementary Fig. 25A) and showed, for the most part, high editing efficiencies (Fig. 2 and Fig. 4). Our observations agree with previous endogenous MRL measurements in human cell lines[46] and differentiating embryonic stem cells[47], where cell type-specific translation effects were mostly driven by 3′UTRs. However, one study found that 5′UTRs may have a diminished regulatory capacity in developing brains compared to cell lines[64]. Therefore, tissue-specific effects may still need to be considered when designing UTRs for certain targets.

To quantify the impact of the very 5′end of the message we then developed MPRAs with shorter, fully randomized 5′UTRs, where only three guanines at the 5′ end are kept constant due to restrictions in T7-based IVT (Fig. 3A). This approach allowed us to observe that out-of-frame uAUGs have a smaller inhibiting effect when located close to the 5′ end (Fig. 3B), whereas poly-C/T tracts have a small enhancing effect that increases with proximity to the 5′ end as well (Fig. 3C, Supplementary Fig. 16). Previous work has observed that multiple types of oligo pyrimidine tracts have a marked effect over translation when located at or near the 5′ end, especially in response to stress or mTOR activation[42–45]. Possibly because of these 5′-proximal regulatory effects, Optimus 5-Prime predictions were unsurprisingly not quite as accurate as for the original libraries which contained a constant 25nt region at the 5′end to facilitate library preparation (Supplementary Fig. 17).

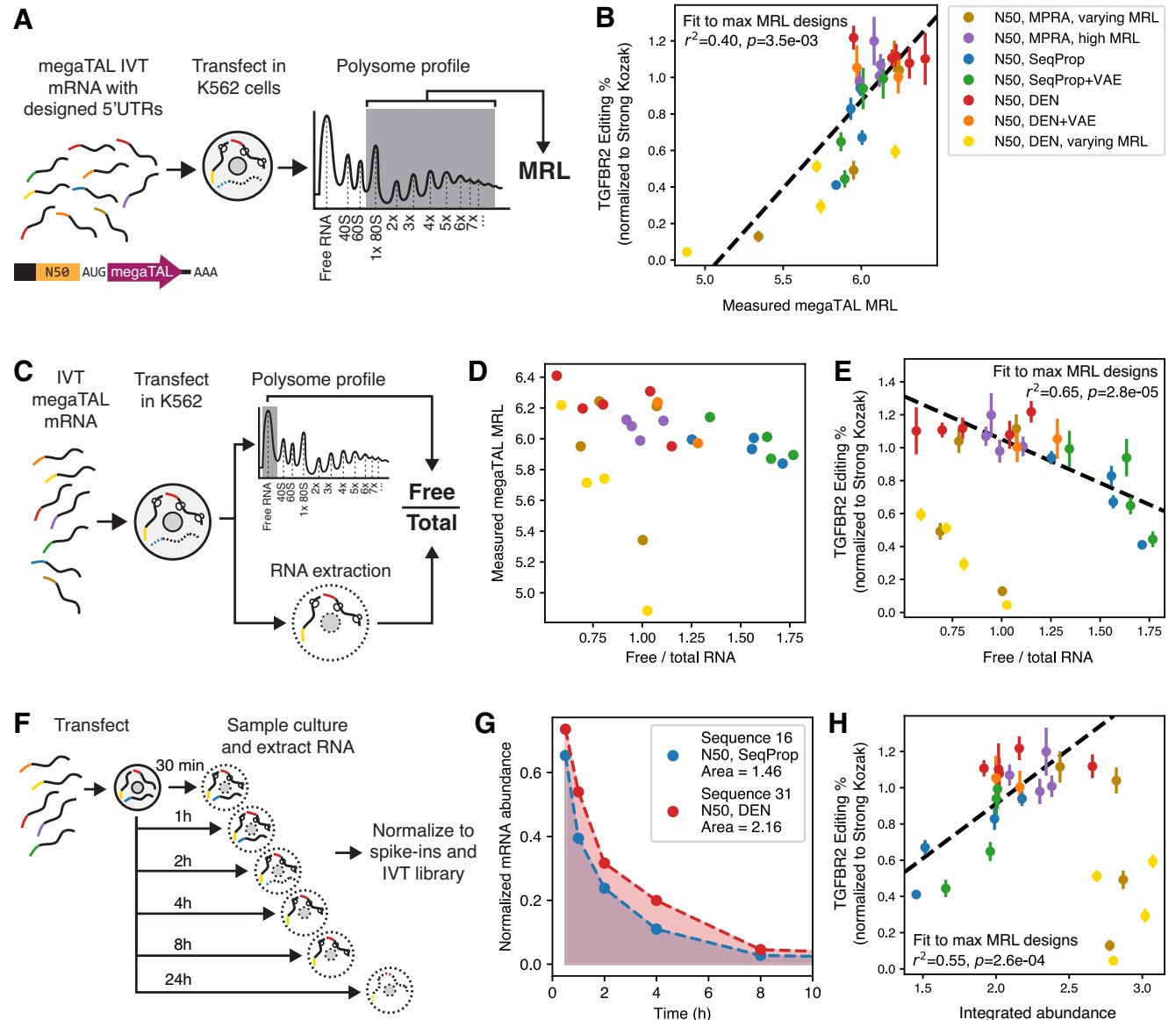

**Fig. 5 | Direct characterization of translation and stability of megaTAL mRNAs explain differences in editing efficiency.** Experiments were performed using megaTAL IVT mRNA containing all designed and selected fixed-end 50nt 5′UTRs from Fig. 2. Dashed regression lines were fitted to data from 5′UTRs designed or selected for maximal MRL (no varying MRL sequences) to assess how these measurements predicted editing efficiency beyond Optimus 5-Prime predicted MRL. **A** Experimental workflow of polysome profiling experiment. **B** *TGFBR2* editing efficiency vs. measured MRL of megaTAL mRNAs. **C** Experimental procedure for quantifying untranslated mRNAs. mRNA abundance in the ribosome-free fraction is measured and normalized to that of the corresponding total cell extract. **D** Comparison of the free-to-total mRNA ratio to measured MRL. **E** *TGFBR2* editing efficiency vs. free-to-total mRNA ratio. **F** Experimental workflow for quantifying the stability of megaTAL mRNAs. mRNA is extracted at different timepoints from transfection and sequenced. Reads are normalized to spike-in controls and to the untransfected IVT library (Methods). **G** mRNA abundance kinetics of two designed 5′UTRs. **H** *TGFBR2* editing efficiency vs. area under the abundance kinetics curve. In (**B**), (**E**), and (**H**), each marker and error bar represent the mean and standard deviation of the Kozak-normalized editing efficiencies, as shown in Fig. 2C, across all mRNA dosages for a particular 5′UTR ($n = 6$ or 8: 3 or 4 dosages across two biological replicates per dosage). p values from a two-sided Wald Test with t-distribution on the test statistic, performed on the linear regression slope with 0 as null hypothesis. No adjustment made for multiple comparisons. Raw UMI counts are provided in Supplementary Data 2. Source data are provided as a Source Data file.

However, we were able to train a new model, Optimus 5-Prime(25), specific to 25 nt-long fully variable 5′UTRs (Fig. 3D, E).

We used model-based design to generate de novo 5′UTRs for strong expression of mRNA-encoded megaTALs, supporting absolute gene editing efficiencies in K562 cells exceeding 40% when targeting the *TGFBR2* gene (Figs. 2C, D and 4A, C) and 80% for *PDCD1* (Supplementary Fig. 11, Supplementary Fig. 23). Notably, most of our designs resulted in high editing efficiencies, matching or exceeding the performance of 5′UTRs taken from the top 0.02% of the random MPRA libraries (Fig. 2C, Fig. 4B). Furthermore, one of our designs resulted in a

*TGFBR2* editing efficiency up to 50% higher than all controls (Fig. 4B), though this effect was not maintained with the *PDCD1*-targeting megaTAL (Fig. 4D, Supplementary Fig. 23). Interestingly, a minimal 5′UTR with only an optimal Kozak sequence achieved editing efficiencies close to the best designed or native 5′UTRs, possibly suggesting that a primary role of naturally occurring 5′UTR sequence-mediated translation regulation is to tune down protein expression to a physiological level. Designs targeting intermediate MRLs also were found to conform to their target values, providing further support for the design approach (Fig. 2E). Together, these results suggest that

model-based design can very reliably maximize a computational predictor. Furthermore, even though we used unmodified RNA, we expect the performance of our designs to be preserved in pseudouridine- and N1-methylpseudouridine-contaning RNA, since we previously showed that these modifications have little effect on MRL[33]. Additionally, gene editing experiments, in agreement with translation MPRAs, showed that the impact of 5'UTR sequences are highly correlated across different cell types, but mechanisms that fully maximize expression may be ORF- and possibly even cell type-dependent. Lu and coworkers recently used a high-throughput assay to test a library of 5'UTRs designed via a random forest model and a genetic algorithm[65], and also observed that the strongest performers varied across CDSs and cell lines. However, their 5'UTRs were placed in the context of a DNA expression cassette and could therefore also influence transcription, making direct comparisons with our results difficult.

Unexpectedly, some Fast SeqProp designs with high predicted MRL resulted in low editing efficiencies (Fig. 2C–G). Furthermore, while VAEs are supposed to help prevent overfitting, they failed to improve the success rate of Fast SeqProp-generated sequences. To directly test whether any of our methods generated poorly predicted sequences or whether the poor performance is due to an effect distinct from MRL, we measured MRLs of all fixed-end 5'UTRs in megaTAL-encoding mRNAs (Fig. 5A, B) and compared them to Optimus 5-Prime predictions. However, we found that prediction error was not substantially larger in Fast SeqProp sequences compared to those designed with DENs or selected from the MPRA (Supplementary Fig. 25A), indicating that all methods were successful at maximizing MRL but that this does not guarantee high functional activity.

To further understand these discrepancies, we investigated other processes that may affect total protein output beyond MRL. For example, our definition of MRL does not include the ribosome-free mRNA fraction. Thus, it is possible, even if counterintuitive, that sequences with high MRL may also have a high fraction of untranslated mRNA, thereby affecting total protein output. Moreover, previous work has shown that mRNA can be destabilized by 5'UTRs with both low[29] and high[28,66] translation efficiencies, which can negatively affect total protein production as well. To better understand these relationships, we quantified mRNA stability and abundance in the free ribosome fraction for a subset of our designs and controls and compared these results to their editing efficiency as a proxy of overall protein abundance (Fig. 5C–H). Focusing on the sequences designed to maximize MRL, we found a positive correlation between editing efficiency and stability (Fig. 5H) and a negative correlation with abundance in the ribosome-free mRNA fraction (Fig. 5E). These correlations are driven by the low-performing Fast SeqProp designs which are notably less stable than higher performing 5'UTRs and more likely to be found in the free mRNA fraction. On the other hand, submaximal designs, which as intended had low and intermediate editing efficiencies, were found to be more stable than successful maximal designs and not particularly enriched in the free ribosome fraction.

We can summarize a few design rules for designing or selecting 5'UTRs that maximize protein output. First, regulatory elements within 5'UTRs seem to mostly repress translation (upstream stop codons, secondary structure, G quadruplexes, etc.). Therefore, even a minimal 5'UTR consisting of a strong Kozak will likely provide a strong initial baseline. As an exception, we found that a T-rich motif at the 5'end has a slightly enhancing effect (Fig. 3C), and accordingly, our algorithms placed such motif in most of our 25nt-long designed 5'UTRs, including the sequence that performed best with the *TGFBR2* megaTAL (Fig. 4). Outside of this initial region, however, additional sequence may risk introducing unintended repressive elements, and thus shorter 5'UTRs should be preferred. Second, 5'UTRs designed or selected to maximize translation may still perform poorly at total protein synthesis due to other mechanisms, the most important of which may be low mRNA stability (Fig. 5F–H). Thus, until the emergence of multi-modal

predictors that capture both translation and stability, model-based 5'UTR design methods that generate diverse sequences, such as DENs, should be preferred to avoid introducing similar artifacts into multiple designs. Third, in most cases, 5'UTRs regulate translation similarly across different cell types (Figs. 1C, 2G and 4E) and CDSs (Figs. 2F and 4D). Even though there might be specific effects in some situations, our results suggest that, for the most part, a strong 5'UTR designed for one context will likely be strong in a different context.

In summary, we used deep learning methods to engineer 5'UTRs that resulted in strong gene expression of a genome editing mRNA therapy. While most of our designs enabled strong gene editing activity, our ability to identify and design outperformers for all ORFs and cell lines of interest is still restricted. By directly measuring translation and stability in a subset of our designs, we were able to identify factors that affect gene editing but are not currently captured by our MRL-based methods. Future work on interrogating the interplay between these phenomena, as well as the interactions between 5UTR, ORF, and 3'UTR sequences constitute promising avenues for further improving UTR design.

## Methods
### Ethical statement
All human PBMC lots were purchased from Key Biologics (currently Charles River), where they were collected from healthy donors under an IRB-approved protocol.

### Polysome profiling in HepG2 cells
HepG2 cells were cultured in EMEM + 10% FBS. The cells were in culture prior to the experiment, passage conditions were 20 mL media and cells at 2e5/mL into a T-75 flask, and cells were allowed to expand for 3 days prior to transfection. Cells used were at passage 6. IVT mRNA corresponding to the 5'UTR 50nt fixed-end library was synthesized during our previous study[33] as follows: Linear DNA containing a T7 promoter, the random 5'UTR, an EGFP CDS, and a truncated BGH poly(A) signal sequence was PCR-amplified from plasmid pET28-IVT-Fixed-N50-EGFP-NheI[33] using primers Pri255 IVT F (GCGAAATT AATACGACTCACTATAGGG, all oligos from IDT) and Pri254_truncBGH_polyT ([T70]CAAACAACAGATGGCTGGCA). A 70nt-long polyA tail was introduced via the reverse PCR primer. IVT was performed using the HiScribe T7 high-yield RNA synthesis kit (NEB E2040), with 3'-O-Me-m7G(5')ppp(5')G RNA cap (NEB S1411) as the cap structure analog. After the reaction, the DNA template was digested with DNase I (NEB M0303) and the IVT mRNA was purified using RNA Clean & Concentrator (Zymo Research R1016). 1 ug IVT mRNA was transfected into one million cells with the Lonza 4D Nucleofector, following the manufacturer's protocol.

Cell lysis was performed 8 h later, followed by polysome profiling, library preparation, and Illumina sequencing as follows[33]: We prepared 10x salt solution (100 mM NaCl, 100 mM MgCl2, 100 mM Tris-HCl pH 7.5 and RNase-free water) and lysis buffer (1 × salt solution, 1% of 20% Triton X-100, 1 mM dithiothreitol, 0.2 U/uL SUPERase-In (Thermo Fisher Scientific AM2694) and 100 μg/mL cycloheximide). For cell lysis, we removed cell media and incubated cells in RNase-free Dulbecco's PBS with 100 ug/mL cycloheximide at 37 C for 5 min, then placed them on ice, washed with cold DPBS with cycloheximide, resuspended in lysis solution, scraped and triturated with a 25-gauge needle, incubated on ice for 10 min, and centrifuged at 16,000 g for 5 min. The supernatant was incubated with 0.005U / uL DNase I (NEB M0303) on ice for 30 min and stored at −80C. For polysome profiling, lysates were placed on top of 20–55% sucrose gradients and centrifuged for 3 h at 151,000 g using a Beckman SW-41 Ti rotor. Fractions corresponding to ribosome peaks were collected and stored at −80C. RNA was purified via trizol/chloroform (Thermo Fisher Scientific 15596018). Reverse transcription was performed using SuperScript IV (Thermo Fisher Scientific 18090200) and

primer Pri289_EGFP_RT (GACGTGTGCTCTTCCGATCT[N10 UMI] AGATGAACTTCAGGGTCAGC). cDNA was amplified with primers AAT GATACGGCGACCACCGAGATCTACAC[8nt i5 index]AGCGTGACAGG GACATCGTAGAGAGTCGTACTTA and CAAGCAGAAGACGGCATACG AGAT[8nt i7 index]GTGACTGGAGTTCAGACGTGTGCTCTTCCGATCT which contain overhangs P5 and P7 for Illumina sequencing and fraction-specific index sequences. Products were sequenced with the Illumina NextSeq platform using NextSeq 500/550 v2 High Output 75 cycle kits. The following custom primers were used: read 1: AGCG TGACAGGGACATCGTAGAGAGTCGTACTTA. Number of cycles were as follows: read 1: 59, read 2: 10, index 1: 8, index 2: 8. Read 1 data should contain variable region of the 5'UTR, whereas read 2 should contain the random UMI.

## Polysome profiling in T cells

T cells were enriched from peripheral blood mononuclear cells (PBMCs) isolated via Ficoll-Paque gradient centrifugation from healthy donors. Activation was performed with anti-CD3/CD28 antibodies in the presence of IL-2. Salt solution and lysis buffer for polysome profiling were prepared as previously described[33]. IVT mRNA corresponding to the 5'UTR 50nt fixed-end library synthesized as part of our previous work[33] was used here as well (see section "Polysome profiling in HepG2"). 1 ug IVT mRNA was transfected into one million cells with the Lonza 4D Nucleofector, following the manufacturer's protocol. Transfected cells were plated in T cell growth medium (TCGM) at 1 million cells/mL and incubated at 37 C, 5% $CO_2$. 8 h later, cyclohex-imide was added dropwise to the media to a final concentration of 100 ug/mL and incubated for 5 additional minutes. Cells were then spined down at 1500 rpm for 5 min and the supernatant was discarded. 300 uL of cold lysis buffer was used to resuspend the cells, and the mixture was incubated on ice for 10 min. Cells were then triturated by passing the mixture through a 25-gauge needle ten times. The mixture was spined down at 16,000 rpm for 5 min at 4 C and the supernatant was transferred to another tube. 1.5 uL of 1U/uL DNAse was added, the mixture was set on ice for 30 min, and stored at −80C. Polysome profiling, library preparation, and Illumina sequencing were performed as described above for HepG2.

## Construction of random-end 5'UTR MPRA libraries

Our previously constructed vector pET28-IVT-Fixed-AgeI-EGFP-NheI[33], containing a T7 promoter followed by the 25nt-long defined 5'UTR prefix and the EGFP CDS, was amplified with primers Bri035 FP EGFP START (TGGGCGAATTAAGTAAGGGC) and Bri042_RP_T7 (CCCT ATAGTGAGTCGTATTAATTTCGCG). This resulted in a linear dsDNA backbone with the complete original vector sequence except for the 25nt-long defined 5'UTR fragment. Random 50nt- and 25nt-long 5'UTRs were introduced by assembling 200 ng of backbone with 10 pmol of primer Bri036_T7_N50_ATG (TAATACGACTCACTATAGGG [N50]ATGGGCGAATTAAGTAAGGG) or Bri043_T7_N25_ATG (GCGAAA TTAATACGACTCACTATAGGG[N25]ATGGGCGAATTAAGTAAGGGCG AGGAGCTGT), respectively, using the NEBuilder HiFi DNA Assembly Master Mix (NEB E2621). 20 uL reactions were incubated at 50 C for 1 h to increase assembly yield. Next, water was added to reach a total volume of 100 uL per reaction, and purification was performed using the column-based DNA Clean & Concentrator 5 (Zymo D4014) with elution in 7uL dH2O. Each library was split in two and transformed separately into 5-alpha cells (NEB, 3.5uL DNA in 35uL cells). IVT template synthesis and IVT were performed as described above for the fixed-end library.

## Polysome profiling of random-end 5'UTR libraries

Transfection, cell lysis, polysome profiling, and RNA extraction were performed as described above for the fixed-end library. Extracted RNA was eluted in 11 uL RNase-free water. Library preparation comprised reverse transcription, template switching, and qPCR amplification, all

of which were performed separately for each polysome fraction. For reverse transcription, we first mixed 10.5 uL of purified RNA with 2 uL of 2 uM primer Bri044_EGFP_RT_primer (AGGGACATCGTAG AGAGTCGTACTTA[N10 UMI]AGATGAACTTCAGGGTCAGC) and 2uL of 10 mM dNTP mixture (NEB N0447), incubated the mixture at 65 C for 5 min, and placed on ice for at least one minute. Then, we added 4 uL Maxima RT buffer, 0.5 uL Superase-In RNAse inhibitor (Thermo Fisher AM2694), and 1 uL Maxima RT RNaseH minus Enzyme (Thermo Fisher EP0751), incubated at 50 C for 15 min, then 85 C for 10 min, and transferred back to ice. Next, we added 1 uL RNase (from bovine pancreas, DNase free, Roche 11119915001) and 1 uL RNase H (NEB M0297) and incubated at 37 C for 15 min. Finally, the product was cleaned with KAPA Pure beads (Roche KK8002) with a 3x ratio of beads to DNA volume, and resuspended in 25 uL dH2O. For template switching, we added 8 uL Maxima RT buffer, 6 uL 50% PEG-8000, 0.5 uL Superase-In, 1 uL Maxima RT RNaseH minus enzyme, and 0.5 uL of 10 uM template switching oligo (AAGCAGTGGTATCAACGCAGAGTA-CATRGrGrG). This mixture was incubated at 42 C for 30 min, then 85 C for 10 min. Next, 1uL RNase (Roche) and 1uL RNase H (NEB) were added and the mixture was incubated at 37 C for 15 min. Finally, the product was cleaned with 3x KAPA Pure beads and resuspended in 20 uL dH2O. qPCR amplification was performed using the KAPA HiFi HotStart ReadyMix (Roche 09420398001) with forward primers AATGATA CGGCGACCACCGAGATCTACAC[N8 index]AGCGTGACAGGGACATC GTAGAGAGTCGTACTTA and reverse primers CAAGCAGAAGACGGC ATACGAGAT[N8 index]AAGCAGTGGTATCAACGCAGA, where the barcodes were specific to each polysome fraction. qPCR reactions were stopped before reaching saturation and purified via gel extractions. Prepared and barcoded libraries corresponding to all polysome fractions of both random-end 25nt replicates and the one 50nt replicate were pooled into a single library for sequencing. Sequencing was performed in an Illumina NextSeq 500 with the NextSeq 500/550 v2 High Output 75 cycle kit. The following custom primers were used: read 1: GCTCCTCGCCCTTACTTAATTCGCCCAT, read 2: CACCTA CGGCAAGCTGACCCTGAAGTTCATCT, index 1: CCCATGTACTCTG CGTTGATACCACTGCTT, index 2: TAAGTACGACTCTCTACGATGTCC CTGTCACGCT. Number of cycles were as follows: read 1: 59, read 2: 10, index 1: 8, index 2: 8. Read 1 data should contain the reverse comple-ment of the variable 5'UTR followed by the reverse complement of the template switching oligo, whereas read 2 should contain the random UMI.

## Processing of polysome profiling sequencing data

Sequencing data from the fixed-end MPRAs in HepG2 cells and T cells (Fig. 1) was processed similarly to our previous HEK293T data[33] as follows: fastq files were generated from the raw instrument output via bcl2fastq with the following options: --no-lane-splitting --minimum-trimmed-read-length 9 --mask-short-adapter-reads 9 --ignore-missing-bcls. We then used cutadapt to retain the first 50nt of read 1. Finally, we clustered UMIs using starcode-umi. For the random-end MPRAs (Fig. 3), we first generated fastq files from the raw instrument output via bcl2fastq as indicated above. We then used a custom python script to retain reads with a mean q-score greater than 25 and where the 3' end of the read 1 sequence matched the expected template switching oligo with a maximum edit distance of 5. We then used starcode-umi to collapse UMIs. For all libraries, we calculated MRL from UMI counts on all polysome fractions as in our previous work[33] by 1) normalizing each fraction library to its total number of reads, 2) summing the product of each fraction's normalized count with their associated number of ribosomes, and 3) dividing by the unweighted sum of normalized counts across all fractions,

## Synthesis of megaTAL mRNA

Ultramers were synthesized encoding the T7 sequence, a 5' UTR, and the first 20 bases of the megaTAL CDS. Template for in vitro

transcription was generated via PCR with the 5' UTR-containing ultra-mers as the forward primer and an ultramer encoding the last 20 bases of the megaTAL CDS and a 125-base polyA tail as the reverse primer. Following plasmid degradation via DpnI, the resulting amplicon was isolated and purified using Ampure beads (Beckman Coulter). In vitro transcription was performed with ARCA co-transcriptional capping. Following DNase treatment to remove residual template, the resulting mRNA was purified using RNase-free Ampure beads. mRNA was run on the Fragment Analyzer (Agilent) to verify expected size and purity, then normalized to 500 nM and stored at −80 C until needed.

### MegaTAL gene editing assay

mRNAs containing fixed-end (Fig. 2) and random-end (Fig. 4) designs, as well as all controls, were assayed simultaneously in the same experiment to facilitate direct comparisons across 5'UTR performance. mRNA at amounts ranging from 2 pmol to 0.25 pmol was electro-porated in duplicate into 100,000 K562 or HepG2 cells per well using the Lonza 4D Nucleofector 96-well shuttle attachment. Electropora-tion conditions were optimized for mRNA transfection of the respec-tive cell type. Following electroporation, cells were cultured at 37 C for 72 h. Cells were lysed in Viagen DirectPCR lysis reagent (cell) following manufacturer protocol to extract gDNA.

### Assessment of gene editing efficiency by amplicon sequencing

Amplification of a ~150 bp region surrounding the megaTAL target site was performed in two PCR reactions. In PCR1, 1.5 uL of genomic DNA was amplified in 30 cycles using gene-specific primers containing Illumina overhangs. In PCR2, P5/P7 sequences and unique combina-tions of i5 and i7 index sequences were appended to yield dual-indexed amplicons in 10 PCR cycles. Samples were pooled, cleaned up with ampure xp beads, and normalized to 16 pM, then run on an Illumina MiSeq with 25% PhiX. BCL data was converted to fastq format and paired ends were merged with PEAR. Reads were demultiplexed and aligned using bowtie2. Editing frequency was calculated as the number of reads that contained insertions or deletions that included part of the 10 base window around the expected breakpoint divided by the total reads with a MAPQ score >20 and quality score >30.

### Polysome profiling and stability measurements of megaTAL mRNAs

mRNAs with all 5'UTRs in Fig. 2 and Fig. 4, with both the *TGFBR2* and *PDCD1* megaTAL CDSs, were pooled together in equal amounts at a total concentration of 500 ng/uL. Both megaTALs were combined to average out CDS effects and only focus on 5'UTR-dependent behavior. K562 cells were grown in RPMI media (Gibco 11875093) with 10% FBS and 1% Penicillin/Streptomycin following standard cell culture meth-ods. For polysome profiling, 20 ug pooled megaTAL mRNA were transfected into 20 million K562 cells using the Neon Electroporation System (Invitrogen MPK5000), transfecting 5 million cells at a time with 5ug mRNA using the 100 uL Neon tip, and otherwise following the manufacturer's instructions. 8 h later, cycloheximide was added to the media at a final concentration of 100 ug/mL. After 10 min at 37 C, cells were centrifuged at 4 C, the supernatant was discarded, and cells were washed twice with DPBS containing 100 ug/mL cycloheximide by resuspension and centrifugation. Cells were then resuspended in 200uL lysis buffer (see the "Polysome profiling in HepG2 cells" section for the composition of the lysis buffer), incubated on ice for 10 min, triturated by passing 10 times through a 25-G needle and syringe, and centrifuged at 4 C. The supernatant was transferred to a new 1.5 mL tube, incubated with 1 unit DNase I (NEB M0303) for 30 min on ice, and stored at −80C. On a different day, the lysate was subject to ultra-centrifugation and fractionation to obtain polysome fractions as described in the "Polysome profiling in HepG2 cells" section. In addi-tion to all ribosome-associated peaks, we collected fractions corre-sponding to the ribosome-free peak. Finally, to quantify total cell

mRNA, 50uL of the lysate was retained and not subjected to polysome profiling. For stability measurements, 5 million K562 cells were trans-fected with 5 ug of pooled megaTAL mRNA following the manu-facturer's instructions, and transferred to 10 mL RPMI media with 10% FBS (no antibiotics). At each timepoint after transfection (30 min, 1 h, 2 h, 4 h, 8 h, 24 h), 1 mL media with cells was extracted, centrifuged at 4 C for 2 min, and carefully washed with cold PBS via resuspension and centrifugation twice to remove residual non-transfected mRNA. Cell pellets were stored at −80C.

Library preparation was performed via a similar procedure as in the "Polysome profiling in HepG2 cells" section, which allowed us to measure abundances of all fixed-end, 50nt 5'UTRs. RNA was extracted from frozen cell pellets or polysome fractions using trizol/chloroform (Thermo Fisher 15596018). To compare mRNA abundances across timepoints, RNA extracts from the stability experiment were spiked in with identical amounts of a mixture of 5 IVT mRNA controls of known sequence, with concentrations proportional to 1, 10, 100, 1,000, and 10,000. RNA extracts were reverse-transcribed using Maxima H Minus RT enzyme (Thermo Fisher EP0753) and an RT primer with sequence GACGTGTGCTCTTCCGATCT[N10 UMI]GGTTTGATCTTCTCTTGCTGC. Samples were then subject to RNase digestion, purified with the DNA Clean & Concentrator (Zymo Research D4014), and eluted in 13uL dH2O. cDNA was amplified with KAPA HiFi master mix (Roche 07958935001) and primers CAAGCAGAAGACGGCATACGAGAT[N8 i7 barcode]GTGACTGGAGTTCAGACGTGTGCTCTTCCGATCT and AATGATACGGCGACCACCGAGATCTACAC[N8 i5 barcode]AGCGTGACAGGGACATCGTAGAGAGTCGTACTT, where barcodes corresponded to polysome fractions or timepoints. 500 ng of the untransfected IVT megaTAL mRNA mixture was processed under an identical protocol starting from the RT step. Libraries were combined and sequenced with an Illumina MiSeq with a MiSeq Nano Kit v2. MRL was calculated from the polysome profiling data as described above. The free-to-total ratio was calculated by dividing the read counts in the ribosome-free frac-tion by those in the total RNA fraction for each sequence.

For kinetics measurements, at each timepoint, a calibration line was fitted to the spike-in controls to relate sequencing reads counts to the most abundant control. This curve was then used to normalize read counts of designed 5'UTRs. A second normalization step was per-formed to the relative read counts in the untransfected IVT library to account for differences in initial loading. Under a simple kinetic model of protein production from mRNA, $\frac{d}{dt}$protein = $r_{translation}$mRNA, protein = $\int_{t_0}^{t_f} mRNA(t)dt$ and therefore the integral of mRNA abun-dance over time should be proportional to megaTAL protein produc-tion. Therefore, we integrated normalized abundances from our first (30 min) to our last (24 h) datapoint, assuming linear changes between timepoints (Fig. 5G).

To perform motif enrichment analysis, we used the MEME tool[67] in differential enrichment mode with the following options: primary sequences: the six sequences with either the largest free-to-total abundance ratio or the lowest stability; control sequences: all remaining assayed sequences; maximum number of motifs to find: 10; minimum width: 5; maximum width: 10; fraction of sequences held out to estimate significance (hsfrac): 0.666666. Only significant hits (E value < 0.05) are reported here.

### Training of Optimus 5-Prime (25)

For every 5'UTR sequence in the random-end 25nt library, a weighted averaged MRL was obtained across replicates, with weights given by the total number of UMI reads per replicate. Sequences were then sorted by read depth, and those with fewer than 100 reads were dis-carded. The top 2000 sequences by read depth were held out for testing, the next 2000 were used for validation/early stopping, and the remaining 193,341 were used for training.

Model training and evaluation were done in Python 3 with ten-sorflow 2. Optimus 5-Prime(25) architecture was based on VGG-16[63]:

it contains a number of convolutional blocks – each with two convolutional layers, one max pooling layer with size and stride 2, and one dropout layer – followed by a fully connected dense layer and a final linear node that computes the MRL. All activations except for the final node are ReLU. The number of convolutional filters in each block is twice the number of filters in the previous block. Models were trained using an MSE loss, and early stopping based on the validation loss was used. Hyperparameter tuning was performed with Amazon Sagemaker using their default Bayesian strategy, with the following parameter ranges: number of convolutional blocks: 1 – 5, kernel size: 2–7, number of filters in the first convolutional block: 16–128, convolutional dropout: 0–0.5, number of units in the final dense layer: 10–100, dropout: 0–0.5. The final architecture is shown in Fig. 3D.

**5′UTR sequence design using Fast SeqProp**

All relevant code was run in Python 3 with keras 2.2 and tensorflow 1.15. Fast SeqProp v0.1 was downloaded from https://github.com/johli/seqprop/. For designs with the 50nt fixed-end architecture (Fig. 2), we used a retrained version of Optimus 5-Prime initially trained on the fixed 50nt MPRA data and finetuned on sequences designed to maximize MRL that contained long poly-U stretches and ultimately underperformed[33]. The loss function to minimize was the sum of a fitness loss plus a sequence loss. The fitness loss was set to the negative of the predicted MRL. The sequence loss was set to the number of occurrences of AUGs across the generated sequence. We also included a term that was set to one if an UG was present at the beginning of the designed region and zero otherwise. This is to penalize an initial uAUG since the last nucleotide of the fixed-end region was A. When VAE regularization was used, an additional term corresponding to the VAE loss was computed by passing the VAE-estimated $p_{VAE}(seq)$ through a margin function $\max(0, margin\_vae - \log(p_{VAE}(seq)))$, where margin_vae = −30. The VAE loss was multiplied by a weight of 0.2 and added to the overall loss function. The number of gradient updates (iterations) was 20,000 for the non-VAE designs and 5000 for the VAE-regularized designs.

Fast SeqProp designs with the 25nt variable-end architecture (Fig. 4) were performed as above, with the following changes: 1) we used Optimus 5-Prime(25) (Fig. 3), 2) we did not include a penalty for an initial UG dinucleotide, 3) margin_vae was set to −15.6, and 4) the vae loss weight was set to 0.4.

**Training of k-mer models for validation of Fast SeqProp designs**

A linear k-mer model was trained on a subset of the HEK293T 50nt fixed-end MPRA data and used as an additional oracle to validate Fast SeqProp designs (Supplementary Fig. 6). Models were trained using the linear_model module of the scikit-learn python package. As training data, MPRA sequences were filtered by discarding those with fewer than 250 reads, those with uAUGs, and those starting with UG to avoid creating an uAUG with the last nucleotide of the fixed 5′ region which was an A. From the remaining 125,931, sequences, 100,000 were used for training and 25,931 for testing. Next, counts of 2, 3, 4, 5, and 6-mers were calculated, and $\log_2(1 + kmer\ counts)$ were computed, resulting in a feature vector of size 5,456. A Lasso model with $\alpha = 0.001$ was trained on a random subset of 50,000 sequences from the training data, resulting in 272 non-zero feature weights. Finally, a Ridge regression model with $\alpha = 0.0$ was trained using only the non-zero features from the Lasso regression model. Performance on the test set is shown in Supplementary Fig. 6B (Pearson r = 0.5213), and model predictions for sequences designed with Fast SeqProp are shown in Supplementary Fig. 6C.

A similar model was trained for validating Fast SeqProp designs with the 25nt random-end architecture (Supplementary Fig. 18). Training sequences were taken from the 25nt random-end MPRA data, then retained only if their read count was greater than 150 and if they did not contain uAUGs. From the remaining 81,552 sequences, 70,000 were used training and 11,552 for testing. Calculation of feature vectors from kmer counts, Lasso regression, and Ridge regression were performed as above. 268 features with nonzero weights resulting from Lasso regression were used for Ridge regression. Performance on the test set is shown in Supplementary Fig. 18A (Pearson r = 0.4094), and model predictions for sequences designed with Fast SeqProp are shown in Supplementary Fig. 18B.

**5′UTR sequence design using Deep Exploration Networks**

All relevant code was run in Python 3 with keras 2.2 and tensorflow 1.15. Deep Exploration Networks (DENs) v0.1 was downloaded from https://github.com/johli/genesis/. In total, we trained five DENs: two for maximizing MRLs in a fixed-end 50nt architecture (Fig. 2), without (Supplementary Fig. 7) and with (Supplementary Fig. 10) VAE regularization, one for designing sequences with submaximal MRLs (inverse regression, Supplementary Fig. 8), and two for maximizing MRLs in a variable-end 25nt architecture (Fig. 4), without (Supplementary Fig. 19) and with (Supplementary Fig. 21) VAE regularization. DEN generator architectures, a short description of the loss function components, and training parameters can be found in the corresponding Supplementary figures.

**Training of Variational AutoEncoders**

VAE training and evaluation were done in Python 3 with keras 2.2 and tensorflow 1.15. Model architectures for the fixed-end 50nt and the random-end 25nt VAEs are shown in Supplementary Fig. 9 and Supplementary Fig. 20 respectively. A high level description of how VAEs are trained and evaluated can be found in Supplementary Fig. 9A, B. Detailed information on the loss function, including equations and derivations for each term, can be found in our previous publication[55]. For the fixed-end 50nt VAE, sequences for training and testing were extracted from our published fixed-end 50nt MPRA dataset[33], by first filtering by read coverage (>2000 reads) and then randomly selecting 5000 (train) and 1000 (test) sequences from the top 10,000 by MRL (-top 25%). For the random-end 25nt VAE, sequences for training and testing were extracted from the random-end 25nt MPRA dataset (Fig. 3), by filtering by read coverage (>500 reads) and then randomly selecting 5000 (train) and 1000 (test) sequences from the top 10,000 by MRL (-top 25%).

**Reporting summary**

Further information on research design is available in the Nature Portfolio Reporting Summary linked to this article.

## Data availability

The polysome profiling data generated in this study (raw fastq files and processed UMI counts) has been deposited in the Gene Expression Omnibus under accession code GSE232927. The polysome profiling data from the fixed-end N50 library from our previous publication[33] can be found in the Gene Expression Omnibus under accession code GSE114002. The processed megaTAL gene editing data is provided in Supplementary Data 1. The processed megaTAL polysome profiling and stability UMI counts are provided in Supplementary Data 2. Source data for every figure in this paper has been deposited to Zenodo [https://doi.org/10.5281/zenodo.11398662].

## Code availability

All code for model training, sequence design, and data analysis has been deposited to github [https://github.com/castillohair/paper-5utr-design] and Zenodo [https://doi.org/10.5281/zenodo.11403014].

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

## Acknowledgements

This work was funded by 2seventy bio, through NSF Award 2021552 and NIH Award R01GM149631 to G.S., and by the University of Washington eScience Institute with support from the Washington Research Foundation to S.C.

## Author contributions

S.C. designed and performed polysome profiling and stability experiments, performed modeling, sequence design, and data analysis, and wrote the manuscript. S.F. designed and performed polysome profiling and gene editing experiments. B.W. designed and performed polysome profiling experiments and data analysis. J.L. performed modeling and sequence design. K.H. designed and performed polysome profiling and gene editing experiments. M.C. designed experiments. G.S. designed experiments and wrote the manuscript.

## Competing interests

SF, KH and MC worked on this project as employees of 2seventy bio. The remaining authors declare no competing interests. GS is a co-founder and shareholder of Parse Biosciences.
