## [Peer Review File · Nature Communications]

Reviewers' Comments:

Reviewer #1:

Remarks to the Author:

Seelig and colleagues report the results of a second generation of their Optimus 5prime deep learning approach to modeling and designing 5' UTR's for increase translation efficiency of synthetic mRNA molecules.

The authors have done an admirably thorough job. The paper fulfills much of a wish list left after the original paper by Sample et al. (Nat. Biotech, 2019) – randomizing the full 5' UTR, making measurements for multiple genes and different cell types, testing a number of deep learning approaches to generate rather than just score designs, and most importantly measuring an actual downstream functional readout, gene editing through a TAL nuclease encoded by the RNA.

Some of the ideas also seem novel for the molecular design field, including use of a VAE to ensure that new neural-network generated sequences are not way out of distribution of the model for translational efficiency. The paper is also clearly written (some suggestions below).

Nevertheless, the authors have been unlucky in that the key features for improving mRNA therapeutics do not seem to involve translational efficiency. Compared to conventional 5' UTRs, most of the approaches tested here do not lead to a positive impact in the final desired result – gene editing efficiency (the one notable boost from a synthetic design does not generalize across cell types or genes). To their credit, the authors state this clearly and have described a number of potential explanations in the Discussion.

One of these explanations is that while increasing translational efficiency can send more ribosomes per unit time down the mRNA, but for reasons that remain mysterious, also decreases the lifetime of the mRNA, so that the total number of protein products created per mRNA ends up roughly constant. (A recent paper from the Chao lab, Dave et al., 2023, Molecular Cell, provides a particularly nice demonstration, which the authors might cite.) So the field/industry as a whole has largely moved to improving mRNA lifetime in cells through changes in, e.g., the coding sequence, which can have strong effects on protein output (see, e.g., Mauger et al., PNAS 2019 from Moderna; Leppek et al., Nat Comms. 2022). The impact of this study's neural network models for the 5' UTR or the special UTR's developed in the paper are unlikely to go into wide use for actual mRNA engineering. However, with some updates, the analysis may be received by the field as a sort of 'final word' for what can(not) be achieved with state-of-the-art 5' UTR design, which could be significant.

1. The main thing that is unclear from the current paper is whether the newly designed 5' UTRs actually increase mean ribosome load or not, and then whether something else then explains the weak effect on gene editing. To really understand what's going on, authors would need to measure mean ribosome load for their designs and ideally mRNA lifetime as well in cells; making both measurements has been a standard in prior work in this area, including the Mauger et al., Leppek et al studies above; and Jia et al., NSMB 2020, cited as ref. 29.
2. Most mRNA engineering studies are using pseudouridine or N1-methyl-pseudouridine, but it doesn't seem that the authors have tested those modifications; the authors should at least give this caveat.
3. I couldn't understand how the model for the 25nt-fully variable UTRs could achieve $r^2 = 0.806$ when the inter-replicate r^2 for those experimental measurements was lower, 0.692.
4. Models and code are available in a GitHub repository, but data are not.
5. Some of the experimental details (IVT, capping, etc) are not described, with a citation to the

prior Sample et al. paper instead. At least a minimal recap would be useful so that readers aren't forced to go back.

Reviewer #2:

Remarks to the Author:

The study from Castillo Hair et al. et al. focuses on AI-guided optimization of the 5' untranslated regions (5'UTRs) of mRNA for increased expression and translation efficiency. Based on their previously developed MPRAs and deep learning methods for both prediction of MRL and sequence generation they conduct experiments using randomized 5'UTR libraries (as compared to fixed 5' end libraries) in different cell types and found that UTR performance was consistent across cell types. They used Optimum 5'Prime (their previous method) trained models on their datasets and employ gradient descent and generative neural networks to guide the design of high-performing 5'UTRs, based also on several tools previously developed by the same authors. They tested some of the designed UTRs with two mRNA encoding gene editing enzymes to target two genes: TGFBR2 and PDCD1. The authors claim that the designed 5'UTRs supported strong gene editing activity, with editing efficiency being consistent across cell types and gene targets. However, the best-performing UTR was specific to one cargo and cell type. The study demonstrates the potential of model-based sequence design for mRNA therapeutics and highlights the importance of optimizing UTRs for enhanced protein expression.

However, for me the main concern of the study is that it is not clear what the focus of it is (what the authors want to demonstrate exactly) and it is also hard to draw some meaningful and definitive conclusions.

The paper does not introduce a novel ML method for 5' UTR design but adapts Optimum 5-Prime (for learning MRL from sequence) across other cell lines and Fast SeqProp and DEN for de novo optimization of 19 5'UTRs (complemented by the use of VAE to optimise sequence design, something the author already showcased in a previous paper). These are great tools, well known to this reviewer, but not clear what the added methodological value of this paper is.

Is the focus of the paper A. the proof of concept that AI-guided 5' UTR design (as done with the machine learning tools previously developed by authors) can be applied to perform efficient gene editing with mega-TAL encoding mRNAs (which is actually nice) or B. find out what is the best design (in terms of sequence reporter length and fixed versus random 5' end, as well insertion of specific motifs) for different applications and different cell lines?

This is equally interesting and important, but In both cases I feel the paper does not provide enough evidence, experiments and/or controls to be able to draw some conclusions. For example:

- The conclusion that 'reporters and methods work in other cell lines' is only partially true. Clearly the HEK293 cell line has an advantage when it come to the ability of Optimum 5-Prime to learn.

Why is that? Is the model learning specific cell-type effects? This should be further discussed.

Evaluation of design performances:

- Is the use of VAE really helping, given than 50% of the Fast seqProp-generated sequences have lower editing efficiencies than the strong Kozak control? Also, the design seems not to work great in the case of HepG2

- Is editing efficiency really a (or the only) measure of 'success'? It correlates with mRNA dosage for all 5'UTR, which is not surprising, but probably at the expenses of some other effects (off target effects, toxicity..). Did the author check that? For therapeutic application this is quite crucial

- The authors try to overcome the limitations of their previous reporter by introducing 25nt or 50 nt random only sequences in their reporter and this is well motivated. But, also there the results are a bit inconclusive, and it is not clear which aspects of the specific application this choice benefits. Is the performance of Optimum 5-Prime ($r^2=0.806$) better, comparable or worse in this setting? Maybe I missed the point here

- The authors design 14 shorter 5' UTRs for megaTAL mRNAs with Fast seqProp and select 4 for gene editing assays. Similarly, by training DEN + VAE, from the top 25 by predicted MRL they select 5. It is not clear how the sequences are selected for validation (more of these examples in the same paragraph). If this is random, one wonders whether the results we see are just random

and what would happen if one was to test the other sequences among the top. Validations should be more extensive to be convincing.

- I am confused why reporters are generated in HEK (and extended to HepG2 and T cells), while gene editing assays are performed in K562. Although the authors show correlations in MRL across cell lines, this is not shown for K562. While this can be easily address, I guess, in addition, there is a (small but present) cell type effect according to their results, this choice of performing validation in K562 (while reporter and models are trained on another cell line) should at the very least be motivated, but in my opinion is penalizing their results and does not sound very coherent.

Minor points:

On page 7 there is a typo 'Supplementary Figure 17!', please correct

Reviewer #3:

Remarks to the Author:

In this manuscript, the authors have modified the deep learning methods in optimizing the 5'UTR. Compared to their previous study (Sample et al., In this manuscript, the authors have modified the deep learning methods in optimizing the 5'UTR. Compared to their previous study (Sample et al., Nature Biotechnology, 2019), the advances in both knowledge and technology in this work are not very strong. My main concerns are as follows:

1. Compared with their previous study, the authors used new mRNA reporter libraries with shorter, 25nt or 50nt-long, 5'UTR. Is there any justification for the determination of the length? In most organisms, the UTRs are mainly over 100bp. Regarding biological relevance, the author should increase the length rather than shorten the length.

2. The authors have modified their previous deep learning method used in their previous study (Sample et al., Nature Biotechnology, 2019) by combining two design methods recently published by the authors (Linder et al., 2020 and 2021). The novelty regarding the methodology in this work is quite weak.

3. Compared to the previous study, the model considers a new factor of uAUG that was not considered in their previous study. It is hard to judge the advances of the modified model in this study.

4. It is not clear how to generalize the design of 5'UTR.

5. It is also not clear how cell types affect the design of 5'UTR.

Reply to the reviewers

Reviewer #1 (Remarks to the Author):

Seelig and colleagues report the results of a second generation of their Optimus 5prime deep learning approach to modeling and designing 5' UTR's for increase translation efficiency of synthetic mRNA molecules.

The authors have done an admirably thorough job. The paper fulfills much of a wish list left after the original paper by Sample et al. (Nat. Biotech, 2019) – randomizing the full 5' UTR, making measurements for multiple genes and different cell types, testing a number of deep learning approaches to generate rather than just score designs, and most importantly measuring an actual downstream functional readout, gene editing through a TAL nuclease encoded by the RNA.

Some of the ideas also seem novel for the molecular design field, including use of a VAE to ensure that new neural-network generated sequences are not way out of distribution of the model for translational efficiency. The paper is also clearly written (some suggestions below).

We thank the reviewer for their supportive comments.

Nevertheless, the authors have been unlucky in that the key features for improving mRNA therapeutics do not seem to involve translational efficiency. Compared to conventional 5' UTRs, most of the approaches tested here do not lead to a positive impact in the final desired result — gene editing efficiency (the one notable boost from a synthetic design does not generalize across cell types or genes). To their credit, the authors state this clearly and have described a number of potential explanations in the Discussion.

One of these explanations is that while increasing translational efficiency can send more ribosomes per unit time down the mRNA, but for reasons that remain mysterious, also decreases the lifetime of the mRNA, so that the total number of protein products created per mRNA ends up roughly constant. (A recent paper from the Chao lab, Dave et al., 2023, Molecular Cell, provides a particularly nice demonstration, which the authors might cite.) So the field/industry as a whole has largely moved to improving mRNA lifetime in cells through changes in, e.g., the coding sequence, which can have strong effects on protein output (see, e.g., Mauger et al., PNAS 2019 from Moderna; Leppek et al., Nat Comms. 2022). The impact of this study's neural network models for the 5' UTR or the special UTR's developed in the paper are unlikely to go into wide use for actual mRNA engineering. However, with some updates, the analysis may be received by the field as a sort of 'final word' for what can(not) be achieved with state-of-the-art 5' UTR design, which could be significant.

This is a good point and we have performed additional experiments to directly measure the mean ribosome load (MRL), the fraction of untranslated (ribosome-free) mRNA, and mRNA stability of a subset of the designed sequences (Fig. 5F-H). While we were successful at

designing 5'UTRs that maximized MRL, we found that their relationship with stability and protein production (as measured by editing efficiency) was somewhat non-linear and not always obvious from prior knowledge.

We observed the best editing efficiencies with 5'UTRs that have very high MRL but do not (fully) maximize stability. Conversely, some sequences with very high stability but intermediate MRL do not result in equally strong editing. When focusing only on sequences with similarly high MRL, we observe a range of stabilities and therefore editing efficiencies. Finally, different proportions of each of these sequences were found to not participate in translation at all, further explaining differences in editing beyond what our current MRL-based methods can predict. In summary, the additional results outline what can be achieved with our state-of-the-art 5'UTR design methods, and point to the interplay between translation, stability, and untranslated mRNA as a promising frontier where additional work has potential to improve UTR design.

A detailed discussion of our experimental findings has been added to a new Section titled "*Differences in the ribosome-free mRNA fraction and in decay kinetics explain performance discrepancies in maximal MRL 5'UTRs designs.*" The Discussion section has been updated as well to account for these results.

We have included a reference to Dave et al. 2023.

1. The main thing that is unclear from the current paper is whether the newly designed 5' UTRs actually increase mean ribosome load or not, and then whether something else then explains the weak effect on gene editing. To really understand what's going on, authors would need to measure mean ribosome load for their designs and ideally mRNA lifetime as well in cells; making both measurements has been a standard in prior work in this area, including the Mauger et al., Leppek et al studies above; and Jia et al., NSMB 2020, cited as ref. 29.

We performed polysome profiling followed by sequencing of fractions to quantify ribosome load for the fixed-end 5'UTRs from our library appended to megaTAL CDS. We also performed a stability time course to determine the influence of the varying 5'UTR sequence on mRNA stability.

The results of the stability and polysome profiling experiments are summarized in **Figs. 5, S25-S27** and the associated text in a new Section titled "*Differences in the ribosome-free mRNA fraction and in decay kinetics explain performance discrepancies in maximal MRL 5'UTRs designs*" and in the Discussion.

Briefly, we find that that measured MRL was generally well correlated with predicted MRL (**Fig. S25A**). However, suboptimal performance of some sequences designed to maximize MRL could be explained by their lower stability and overrepresentation in the ribosome-free fraction in the polysome profile (**Fig. 5C-H**).

2. Most mRNA engineering studies are using pseudouridine or N1-methyl-pseudouridine, but it doesn't seem that the authors have tested those modifications; the authors should at least give this caveat.

This is a good point and the following sentence is include in the discussion:

“Furthermore, even though we used unmodified RNA, we expect the performance of our designs to be preserved in pseudouridine- and N1-methyl pseudouridine-containing RNA, since we previously showed that these modifications have little effect on MRL (Sample, et al. 2019)”

3. I couldn't understand how the model for the 25nt-fully variable UTRs could achieve $r^2 = 0.806$ when the inter-replicate r^2 for those experimental measurements was lower, 0.692.

The apparent discrepancy is a result of using different sets of sequences to calculate both values. To calculate the inter-replicate correlation, we used a set of 20,000 5'UTRs, a number chosen to allow the r^2 to be compared to our previous library and publication.

In the “Measuring translation efficiency of short, fully variable 5'UTRs” section, we write:

*“Inter-replicate MRL correlation was good ($r^2 = 0.692$ for the top 20,000 sequences by read coverage, **Supplementary Figure 13C, D**), although lower than our previous “fixed-end” 50nt library ($r^2 = 0.938$, **Supplementary Figure 1D**).”*

In contrast, r^2 for model predictions was achieved on a higher-quality held-out test set of 2,000 5'UTRs with the highest read coverage. All other UTRs in the library were used for model training, with the rationale of having the largest possible training set while keeping the test set reliable. Inter-replicate r^2 for these sequences is 0.844, which is indicated at the beginning of the section “Predicting translation efficiency from short, fully variable 5'UTRs”:

*“We evaluated candidate models via their prediction accuracy on the top 2,000 sequences by read coverage from the random-end 25nt MPRA library, which showed good inter-replicate correlation ($r^2 = 0.844$, **Supplementary Figure 13D**).”*

Further down in the text, when discussing the model trained on 25nt-variable UTR dataset we updated the text to clarify the test-set used:

*“This model showed good performance on the same test set which was held out from training ($r^2 = 0.806$ on the top 2,000 sequences by read coverage, **Figure 3E**).”*

We have also included plots showing that when sequences with the highest quality are used to calculate inter-replicate correlation, increasing the number of sequences results in an apparent drop in r^2 as sequences with decreasing quality are included. (Supplementary Figure 1D, Supplementary Figure 13D).

4. Models and code are available in a GitHub repository, but data are not?

All data are submitted to GEO and an accession number is provided in the manuscript. A pointer was added to the GitHub repository.

5. Some of the experimental details (IVT, capping, etc) are not described, with a citation to the prior Sample et al. paper instead. At least a minimal recap would be useful so that readers aren't forced to go back.

We have added a more detailed description of our IVT and polysome profiling procedures to the "Polysome profiling in HepG2 cells" subsection of the methods section, and referenced this in subsequent subsections when necessary.

Reviewer #2 (Remarks to the Author):

The study from Castillo Hair et al. et al. focuses on AI-guided optimization of the 5' untranslated regions (5'UTRs) of mRNA for increased expression and translation efficiency. Based on their previously developed MPRAs and deep learning methods for both prediction of MRL and sequence generation they conduct experiments using randomized 5'UTR libraries (as compared to fixed 5' end libraries) in different cell types and found that UTR performance was consistent across cell types. They used Optimum 5'Prime (their previous method) trained models on their datasets and employ gradient descent and generative neural networks to guide the design of high-performing 5'UTRs, based also on several tools previously developed by the same authors. They tested some of the designed UTRs with two mRNA encoding gene editing enzymes to target two genes: TGFBR2 and PDCD1. The authors claim that the designed 5'UTRs supported strong gene editing activity, with editing efficiency being consistent across cell types and gene targets. However, the best-performing UTR was specific to one cargo and cell type. The study demonstrates the potential of model-based sequence design for mRNA therapeutics and highlights the importance of optimizing UTRs for enhanced protein expression.

We thank the reviewer for these encouraging comments.

However, for me the main concern of the study is that it is not clear what the focus of it is (what the authors want to demonstrate exactly) and it is also hard to draw some meaningful and definitive conclusions.

The paper does not introduce a novel ML method for 5' UTR design but adapts Optimum 5-Prime (for learning MRL from sequence) across other cell lines and Fast SeqProp and DEN for de novo optimization of 19 5'UTRs (complemented by the use of VAE to optimise sequence design, something the author already showcased in a previous paper). These are great tools, well known to this reviewer, but not clear what the added methodological value of this paper is.

Is the focus of the paper A. the proof of concept that AI-guided 5' UTR design (as done with the machine learning tools previously developed by authors) can be applied to perform efficient gene editing with mega-TAL encoding mRNAs (which is actually nice) or B. find out what is the best design (in terms of sequence reporter length and fixed versus random 5' end, as well insertion of specific motifs) for different applications and different cell lines?

We believe that our work makes progress towards both goals: We show that synthetic 5'UTRs can drive efficient gene editing and also define design parameters and approaches that increase

the success rate of 5'UTR design for future applications. Specifically, our work makes the following contributions:

- We compare the performance of a large 5'UTR library in three different cell types including primary activated T cells relevant for cell therapy applications.
- We measure and model the impact of shorter UTRs and of variation targeted to the very 5'end of the message.
- We functionally test 5'UTRs in the context of megaTAL gene editing enzymes demonstrating the potential of synthetic 5'UTRs for practical applications. We compare gene editing efficiency across two cell lines and gene targets.
- We perform stability and ribosome loading experiments to explain discrepancies between the predicted MRL and observed editing performance for a subset of sequences.

To address point B and summarize our results in the form of a recommendation for future applications, we added the following paragraph to the Discussion:

*“We can summarize a few design rules for designing or selecting 5'UTRs that maximize protein output. First, regulatory elements within 5'UTRs seem to mostly repress translation (upstream stop codons, secondary structure, G quadruplexes, etc.). As an exception, we found that a T-rich motif at the 5'end has a slightly enhancing effect (**Figure 3C**), and accordingly, our algorithms placed such motif in most of our 25nt-long designed 5'UTRs, including the sequence that performed best with the TGFBR2 megaTAL (**Figure 4**). Outside of this initial region, however, additional sequence may risk introducing unintended repressive elements, and thus shorter 5'UTRs should be preferred. Second, 5'UTRs designed or selected to maximize translation may still perform poorly at total protein synthesis due to other mechanisms, the most important of which may be low mRNA stability (**Figure 5F-H**). Thus, until the emergence of multi-modal predictors that capture both translation and stability, model-based 5'UTR design methods that generate diverse sequences, such as DENs, should be preferred to avoid introducing similar artifacts into multiple designs. Third, in most cases, 5'UTRs regulate translation similarly across different cell types (**Figure 1C**, **Figure 2G**, **Figure 4E**) and CDSs (**Figure 2F**, **Figure 4D**). Even though there might be specific effects in some situations, our results suggest that, for the most part, a strong 5'UTR designed for one context will likely be strong in a different context.”*

This is equally interesting and important, but In both cases I feel the paper does not provide enough evidence, experiments and/or controls to be able to draw some conclusions. For example:

- The conclusion that 'reporters and methods work in other cell lines' is only partially true. Clearly the HEK293 cell line has an advantage when it come to the ability of Optimum 5-Prime to learn. Why is that? Is the model learning specific cell-type effects? This should be further discussed.

We believe that the better performance observed with HEK cells is primarily a data quality issue and not necessarily a biological phenomenon. Because HEK cells are easy to transfect and grow rapidly, we were able to deliver more RNA (14.5 ug vs 1 ug) and use a larger number of cells (>4 million on transfection day vs 1 million) for the experiments performed in the original publication, resulting in exceptionally high quality data. You can observe a similar difference in the data correlation across replicates for HEK293T ($r^2 = 0.938$) versus T cells ($r^2 = 0.814$), which supports our hypothesis that these differences are driven mostly by data quality.

To further clarify these points, we added the following sentences to the section “Optimus 5-Prime predictions generalize to cells relevant to mRNA therapeutics”:

In the second paragraph, when discussing the MRL data quality of HepG2 and T cells:

“While these were lower than the HEK293T inter-replicate correlation ($r^2 = 0.938$), this difference could be at least partially explained by the higher HEK293T data quality resulting from a larger number of cells (> 4 million vs 1 million) and amount of IVT RNA (14.5 ug vs 1 ug) used in our previous study”

In the next paragraph when describing the performance of our original Optimus 5-Prime:

“For both HEK293T and T cells, performance was close to inter-replicate correlation, thus absolute differences in r^2 are likely due to the higher data quality in HEK293T.”

Evaluation of design performances:

- Is the use of VAE really helping, given that 50% of the Fast seqProp-generated sequences have lower editing efficiencies than the strong Kozak control? Also, the design seems not to work great in the case of HepG2

The dataset is not large enough to conclusively determine the utility of VAEs for this application but based on our observations, they do not appear to be helpful.

Regarding the lower performance of some of the Fast SeqProp designs, we have performed additional experiments, where we directly measure the MRL and stability of the designed sequences with the goal of explaining the observed performance variation beyond predicted MRL. The results of these experiments are summarized in **Fig. 5, Supplementary Figs. S25-S27** and the associated text in a new Section titled “*Differences in the ribosome-free mRNA fraction and in decay kinetics explain performance discrepancies in maximal MRL 5'UTRs designs*” and in the Discussion.

Briefly, we find that that measured MRL was generally well correlated with predicted MRL, suggesting that the methods (with or without VAE) successfully maximize MRL (**Fig. S25A**). However, suboptimal performance of some sequences designed to maximize MRL could be explained by their lower stability and overrepresentation in the ribosome-free fraction in the polysome profile (**Fig. 5C-H**).

- Is editing efficiency really a (or the only) measure of 'success'? It correlates with mRNA dosage for all 5'UTR, which is not surprising, but probably at the expenses of some other effects (off target effects, toxicity..). Did the author check that? For therapeutic application this is quite crucial

We have not characterized off-target editing in this work, however, the megaTALs have been extensively characterized in-house at 270bio.

- The authors try to overcome the limitations of their previous reporter by introducing 25nt or 50 nt random only sequences in their reporter and this is well motivated. But, also there the results are a bit inconclusive, and it is not clear which aspects of the specific application this choice benefits. Is the performance of Optimum 5-Prime ($r^2=0.806$) better, comparable or worse in this setting? Maybe I missed the point here

We had two motivations in studying the impact of shorter 5'UTR and randomization closer to the 5'UTR. First, we aimed to understand the contribution of the very 5' end of the message to ribosome loading. We write:

"5'UTR regulation may differ when sequence elements are placed close to the 5' terminus. For example, various pyrimidine-rich motifs have been found to influence translation in response to stress when located within a few bases of the 5' end. Our previous 5'UTR MPRA was unable to interrogate this region, as a fixed 25nt segment was placed at the 5' end to facilitate library preparation."

We indeed observe some sequence motifs with varying impact as a function of the distance from the start and proximity to the 5' cap. These results are shown in **Fig. 3** and discussed in the corresponding main paper text.

A second, more practical motivation was to ask whether we could reduce the size of the 5'UTR without reducing its ability to load ribosomes. If nothing else, shorter sequence elements make it less likely that information is encoded that will result in an unexpected response in cell types or contexts not seen during training. Moreover, in particular in the context of AAV-based gene therapy, it may be preferable to reduce the size of the therapeutic cargo.

- The authors design 14 shorter 5' UTRs for megaTAL mRNAs with Fast seqProp and select 4 for gene editing assays. Similarly, by training DEN + VAE, from the top 25 by predicted MRL they select 5. It is not clear how the sequences are selected for validation (more of these examples in the same paragraph). If this is random, one wonders whether the results we see are just random and what would happen if one was to test the other sequences among the top. Validations should be more extensive to be convincing.

Sequences for testing were selected randomly from the designs. Our main bottleneck was the comparably limited throughput of the gene editing experiments which need to be performed in an arrayed rather than pooled format.

We note that the vast majority of our designs work as expected. The primary exceptions are a subset of the sequences designed with Fast SeqProp to maximize ribosome load which result in sub-maximal gene editing.

We have performed more extensive validation experiments to understand why this subset of sequences did not perform as expected. Specifically, we directly measured the MRL and stability of the designed sequences. The results of these experiments are summarized in **Figs. 5, S25-S27** and the associated text in a new Section titled “*Differences in the ribosome-free mRNA fraction and in decay kinetics explain performance discrepancies in maximal MRL 5’UTRs designs*” and in the Discussion.

Briefly, we find that that measured MRL was generally well correlated with predicted MRL, suggesting that all design methods successfully maximize MRL (**Fig. S25A**). However, suboptimal performance of some sequences designed to maximize MRL could be explained by their lower stability and overrepresentation in the ribosome-free fraction in the polysome profile (**Fig. 5C-H**).

- I am confused why reporters are generated in HEK (and extended to HepG2 and T cells), while gene editing assays are performed in K562. Although the authors show correlations in MRL across cell lines, this is not shown for K562. While this can be easily address, I guess, in addition, there is a (small but present) cell type effect according to their results, this choice of performing validation in K562 (while reporter and models are trained on another cell line) should at the very least be motivated, but in my opinion is penalizing their results and does not sound very coherent.

Initial data was generated in HEK cells because of the ease of cell culture and highly efficient delivery. Motivated by applications targeting the liver and immune system, we also tested a subset of our libraries in HepG2 and T cells. We decided to use K562 rather than T cells for gene editing experiments because of the lower cost and because they are easier to work with, but also because genetically modified K562 are used to induce activation and expansion of engineered natural killer and CAR-T cells.

It is true that the correlation between different cell types is not perfect and we observe some degree of specificity. Still, a major result from our experiments is that 5’UTR mediated translation regulation is broadly similar across different cell types and that UTRs designed in one context should still be (highly) functioning in other contexts. This is further supported by our new measurements, where we find that the experimental MRLs are highly correlated with Optimus 5-Prime predictions (**Supplementary Figure 25A**), but other effects, such as mRNA stability, drive some of the observed performance differences (**Figure 5**). We have added the following sentences to the discussion to emphasize this point: “...our results suggest that synthetic 5’UTRs regulate translation from IVT mRNA with little specificity across human cells relevant for mRNA therapeutics. This is further supported by our editing results: sequences designed using a model trained on HEK293T data resulted in MRLs in K562 cells that were highly correlated with predictions (**Supplementary Figure 25A**) and showed, for the most part, high editing efficiencies (**Figure 2 and Figure 4**).”

Minor points:

On page 7 there is a typo 'Supplementary Figure 17!', please correct

We have corrected this mistake.

Reviewer #3 (Remarks to the Author):

In this manuscript, the authors have modified the deep learning methods in optimizing the 5'UTR. Compared to their previous study (Sample et al., In this manuscript, the authors have modified the deep learning methods in optimizing the 5'UTR. Compared to their previous study (Sample et al., Nature Biotechnology, 2019), the advances in both knowledge and technology in this work are not very strong.

We believe that there are several important and interesting results that go beyond our earlier work. The primary novel points in the paper are:

- We compare the performance of a large 5'UTR library in three different cell types including primary activated T cells relevant for cell therapy applications.
- We measure and model the impact of shorter UTRs and of variation targeted to the very 5'end of the message.
- We functionally test 5'UTRs in the context of megaTAL gene editing enzymes demonstrating the potential of synthetic 5'UTRs for practical applications. We compare gene editing efficiency across two cell lines and gene targets.
- We perform stability and ribosome loading experiments to explain discrepancies between the predicted MRL and observed editing performance for a subset of sequences.

My main concerns are as follows:

1. Compared with their previous study, the authors used new mRNA reporter libraries with shorter, 25nt or 50nt-long, 5'UTR. Is there any justification for the determination of the length? In most organisms, the UTRs are mainly over 100bp. Regarding biological relevance, the author should increase the length rather than shorten the length.

We had two motivations in studying the impact of shorter 5'UTR and randomization closer to the 5'UTR. First, we aimed to understand the contribution of the very 5'end of the message to ribosome loading. We write:

"5'UTR regulation may differ when sequence elements are placed close to the 5' terminus. For example, various pyrimidine-rich motifs have been found to influence translation in response to stress when located within a few bases of the 5' end. Our previous 5'UTR MPRA was unable to interrogate this region, as a fixed 25nt segment was placed at the 5' end to facilitate library preparation."

We indeed observe some sequence motifs with varying impact as a function of the distance from the start and proximity to the 5' cap. These results are shown in **Fig. 3** and discussed in the corresponding main paper text.

A second, more practical motivation was to ask whether we could reduce the size of the 5'UTR without reducing its ability to load ribosomes. If nothing else, shorter sequence elements make it less likely that information is encoded that will result in an unexpected response in cell types or contexts not seen during training. Moreover, in particular in the context of AAV-based gene therapy, it may be preferable to reduce the size of the therapeutic cargo. We should point out that the choice of shortening the 5'UTR was not made to more closely resemble endogenous 5'UTRs, as longer 5'UTRs, including random and endogenous sequences were characterized in our previous study.

2. The authors have modified their previous deep learning method used in their previous study (Sample et al., Nature Biotechnology, 2019) by combining two design methods recently published by the authors (Linder et al., 2020 and 2021). The novelty regarding the methodology in this work is quite weak.

We do not claim novelty with respect to the design or deep learning algorithms. Instead, our goal was to use state of the art algorithms to generate sequences for functional testing in the context of gene editing. We have added the following line to the introduction:

“In this study, we designed de novo 5'UTRs for an mRNA-encoded gene editing application using Optimus 5-Prime, thereby validating our previously established deep learning modeling and sequence design methods in the context of a functional assay.”

3. Compared to the previous study, the model considers a new factor of uAUG that was not considered in their previous study. It is hard to judge the advances of the modified model in this study.

Our previous work has observed and modeled the importance of uAUGs in the context of translation regulation. Relevant data are shown, for example, in Fig. 1C in Sample et al. NBT 37, 803 (2019) and Fig. 2 of Cuperus et al. Genome Res. 27, 2015 (2017).

We first discuss the impact of uAUGs in the current paper to validate our measurements. Out-of-frame uAUGs are expected to have a strong negative impact and this effect is clearly visible in our experiments. Second, we also observe a pronounced dependency on the distance of the AUG from the cap in our varying-end libraries which we could not observe in our earlier assays and which are captured by the model. Unrelated to AUGs, we also observe an enhancing effect from short U-rich sequences near the 5'cap which we could not characterize in our previous work. Therefore, our new model was trained to capture these effects specific to the 5' end of the mRNA.

5. It is also not clear how cell types affect the design of 5'UTR.

Our main observations from performing MPRA in multiple cell types are detailed in the Section titled “*Optimus 5-Prime predictions generalize to cells relevant to mRNA therapeutics*”. In short, we found that MRLs from our large library of randomized 5’UTRs were highly correlated across HEK293T, HepG2, and T cells. Furthermore, Optimus 5-Prime predictions are highly predictive of measured MRLs in all cell lines, with prediction-measurement correlations close to inter-replicate correlations. Additionally, models trained specifically on HepG2 and T cell measurements did not perform better than the original model trained on HEK293T, suggesting that there is very little cell type-specific information in the data.

Finally, this general conclusion is further supported by our gene editing results: 5’UTRs designed with a model trained on HEK293T were tested in K562 cells and led to high gene editing efficiency in most cases (Figure 2, Figure 4) and experimentally measured MRLs that were highly correlated with predictions (Supplementary Figure 25A). To emphasize these points, we have added the following to the first paragraph of the Discussion section: “*Therefore, our results suggest that synthetic 5’UTRs regulate translation from IVT mRNA with little specificity across human cells relevant for mRNA therapeutics. This is further supported by our gene editing results: sequences designed using a model trained on HEK293T data resulted in MRLs in K562 cells that were highly correlated with predictions (Supplementary Figure 25A) and showed, for the most part, high editing efficiencies (Figure 2 and Figure 4).*”

Reviewers' Comments:

Reviewer #1:

Remarks to the Author:

The authors have satisfactorily addressed my comments.

Reviewer #2:

Remarks to the Author:

The authors have addressed all my points in the revised version of the manuscript. The points of concerns have been addressed with further discussion or additional explanations. I particularly appreciate that they extended their validation experiments to confer robustness to their conclusions. I enjoyed this manuscript and have no additional remaining concerns on my side.

Reviewer #3:

Remarks to the Author:

In the revised manuscript, the authors have clarified the design purpose and provided more interpretations of their results. However, I still have several concerns, as follows:

Comment on the response to Comment 1.

While I appreciate the authors' efforts in clarifying their work, I still have some points. Firstly, based on Figures 2 and 4, it is challenging to conclude that the design of a shorter 5'UTR outperformed the design of a fixed and longer 5'UTR (25+50). Secondly, the authors should comprehensively characterize and compare the designed 5'UTR with the model-based designs using the shorter and longer 5'UTR. Lastly, it is crucial to determine the requirement of certain features, such as length limitation and sequencing content, for designing a functional and efficient 5'UTR. This work aims to provide an optimizing strategy in designing 5'UTR for translation, and it is important that the authors provide the "design rule".

Comment on the response to Comment 2.

While I appreciate the authors' clarification, if the focus is on the application for designing an efficient 5'UTR, we should expect more efficient 5'UTRs than the Kozak sequence. It feels that the 6 nt upstream AUG (Kozak) could reach the best 5'UTR design. Therefore, the authors should provide more robust evidence to support the performance of the model-based design of 5'UTR.

Comment on the response to Comment 3.

The model incorporated the information of uAUGs and short pyrimidine-rich motifs into the design. However, the pyrimidine-rich motifs did not appear in most of the design molecules. If the current model-based design on 25nt libraries provides the pyrimidine-rich motif requirement close to 5'end, the derived designs with pyrimidine-rich should be the outperformed candidates. If not the pyrimidine-rich rule, are there any other features that the authors could extract from their model? Again, the authors should offer some novel insights.

Reviewer #1 (Remarks to the Author):

The authors have satisfactorily addressed my comments.

Reviewer #2 (Remarks to the Author):

The authors have addressed all my points in the revised version of the manuscript. The points of concerns have been addressed with further discussion or additional explanations. I particularly appreciate that they extended their validation experiments to confer robustness to their conclusions. I enjoyed this manuscript and have no additional remaining concerns on my side.

Reviewer #3 (Remarks to the Author):

In the revised manuscript, the authors have clarified the design purpose and provided more interpretations of their results. However, I still have several concerns, as follows:

Comment on the response to Comment 1.

While I appreciate the authors' efforts in clarifying their work, I still have some points. Firstly, based on Figures 2 and 4, it is challenging to conclude that the design of a shorter 5'UTR outperformed the design of a fixed and longer 5'UTR (25+50).

In practice, the data shown in Figs. 2 and 4 was collected in the same experiment performed in a 96-well format. The data can therefore be compared directly between the two figures, and one can conclude that the DEN-designed shorter 5'UTR outperformed all others when used with the TGFBR2 megaTAL.

We made the following modifications in the text to emphasize this point:

1) At the beginning of the "Designed short 5'UTRs enable high megaTAL-induced gene editing activity" section: "We then evaluated the gene editing performance of these designs in K562 as above (**Figure 2B**). In practice, we assayed "defined-end" (**Figure 2C-G**) and "random-end" (**Figure 4**) sequences simultaneously to enable direct comparisons across 5'UTR architectures (**Methods**)."

2) Later in the same paragraph (only the phrase in italics is new): "Moreover, when targeting the TGFBR2 gene, one DEN-designed sequence outperformed all other UTRs, *including defined-end designs*, at all mRNA dosages (absolute efficiency of 55.6% at 2 pmol mRNA), improving over the previous best performer LAMA5 by 18-33% (Figure 4C)."

3) The methods section: "mRNAs containing fixed-end (**Figure 2**) and random-end (**Figure 4**) designs, as well as all controls, were assayed simultaneously in the same experiment to facilitate direct comparisons across 5'UTR performance."

Secondly, the authors should comprehensively characterize and compare the designed 5'UTR with the model-based designs using the shorter and longer 5'UTR.

We would like to emphasize that all designed 5'UTRs in this manuscript are model-based designs. As for characterization across short and long 5'UTRs, we have performed gene editing experiments with both sets of UTRs simultaneously, as we describe above, therefore their gene editing performances can be directly compared. We should additionally note that we are not claiming that the designed 5'UTRs perform best in all contexts with all CDSs. In fact, we found that, while a synthetic UTR outperformed all others with the TGFBR2 megaTAL, an endogenous UTR was better with the PDCD1 megaTAL. Fully characterizing the interplay between CDS and UTR sequences is outside the scope of this work, but until this is done, the best UTR

characterization one can perform for a particular application is with the relevant CDS measuring the functional readout of interest. This is what we do in Figures 2 and 4, by placing UTRs upstream of megaTAL gene editors and measuring gene editing directly. As noted above, we perform these experiments with replicates across *four different mRNA dosages* to ensure our results and comparisons across UTRs are robust for the application of megaTAL gene editing.

Lastly, it is crucial to determine the requirement of certain features, such as length limitation and sequencing content, for designing a functional and efficient 5'UTR. This work aims to provide an optimizing strategy in designing 5'UTR for translation, and it is important that the authors provide the "design rule".

This is a good point and we have added a summary of our design recommendations to the Discussion section in response to a similar comment by Reviewer 2 in the previous revision. Specifically, we wrote:

"We can summarize a few design rules for designing or selecting 5'UTRs that maximize protein output. First, regulatory elements within 5'UTRs seem to mostly repress translation (upstream stop codons, secondary structure, G quadruplexes, etc.). As an exception, we found that a T-rich motif at the 5'end has a slightly enhancing effect (**Figure 3C**), and accordingly, our algorithms placed such motif in most of our 25nt-long designed 5'UTRs, including the sequence that performed best with the TGFBR2 megaTAL (**Figure 4**). Outside of this initial region, however, additional sequence may risk introducing unintended repressive elements, and thus shorter 5'UTRs should be preferred. Second, 5'UTRs designed or selected to maximize translation may still perform poorly at total protein synthesis due to other mechanisms, the most important of which may be low mRNA stability (**Figure 5F-H**). Thus, until the emergence of multi-modal predictors that capture both translation and stability, model-based 5'UTR design methods that generate diverse sequences, such as DENs, should be preferred to avoid introducing similar artifacts into multiple designs. Third, in most cases, 5'UTRs regulate translation similarly across different cell types (**Figure 1C**, **Figure 2G**, **Figure 4E**) and CDSs (**Figure 2F**, **Figure 4D**). Even though there might be specific effects in some situations, our results suggest that, for the most part, a strong 5'UTR designed for one context will likely be strong in a different context."

Comment on the response to Comment 2.

While I appreciate the authors' clarification, if the focus is on the application for designing an efficient 5'UTR, we should expect more efficient 5'UTRs than the Kozak sequence. It feels that the 6 nt upstream AUG (Kozak) could reach the best 5'UTR design. Therefore, the authors should provide more robust evidence to support the performance of the model-based design of 5'UTR.

It is indeed interesting that a minimal 5'UTR with only a strong Kozak sequence can drive strong gene editing. However, our measurements also show that at least in the context of the MegaTAL CDSs tested here, the Kozak sequence alone does not reach the performance of the best 5'UTR. It should be noted that we are not claiming that a single one of our synthetic 5'UTRs will perform best in all contexts and CDSs. In fact, a synthetic UTR was optimal for TGFBR2 megaTAL gene editing, but a natural one (VAT1) was optimal for the PDCD1 megaTAL, and both showed higher gene editing than the minimal 5'UTR in each case. Therefore, until these CDS-dependent effects are characterized more comprehensively in a future study, the best characterization one can perform is to measure 5'UTR performance with their CDS of interest using the relevant functional readout. This is, in fact, what we do: we demonstrate the relationships described above by placing 5'UTRs upstream of megaTAL CDSs and measuring gene editing across four different mRNA dosage levels and in two replicates (Figure 2C-D, Figure 4A-C).

Nevertheless, we agree with the reviewer that the minimal 5'UTR has a remarkably good performance. We have made the following additions to the discussion section to acknowledge this more explicitly:

- 1) Paragraph 3: we added “Interestingly, a minimal 5’UTR with only an optimal Kozak sequence achieved editing efficiencies close to the best designed or native 5’UTRs, possibly suggesting that a primary role of naturally occurring 5’UTR sequence-mediated translation regulation is to tune down protein expression to a physiological level”
- 2) Paragraph 6, where we summarize our design rules: “Therefore, even a minimal 5’UTR consisting of a strong Kozak will likely provide a strong initial baseline”

Comment on the response to Comment 3.

The model incorporated the information of uAUGs and short pyrimidine-rich motifs into the design. However, the pyrimidine-rich motifs did not appear in most of the design molecules. If the current model-based design on 25nt libraries provides the pyrimidine-rich motif requirement close to 5'end, the derived designs with pyrimidine-rich should be the outperformed candidates. If not the pyrimidine-rich rule, are there any other features that the authors could extract from their model? Again, the authors should offer some novel insights.

We thank the reviewer for noticing that we did not point out the importance of the pyrimidine-rich motif in the context of the designed 5’UTRs. In fact, we do observe that many of the synthetic 5’UTRs contain such motifs. Within the designed 25nt 5’UTRs that were characterized in gene editing experiments, 5 out of 7 Fast SeqProp and 4 out of 7 DEN-generated sequences contain TTY following the IVT-obligatory GGG, even though sequences were optimized based on predicted MRL and not explicitly on any sequence features. We have added the following sentence to the manuscript to highlight this point:

“Notably, despite optimizing based on MRL only, our designed sequences predominantly started with an oligopyrimidine repeat: from our final selection, 5 out of 7 Fast SeqProp and 4 out of 7 DEN-generated sequences contain TTY after the initial GGG required by IVT, indicating that our methods captured and exploited the enhancing effect of this sequence motif (**Figure 3C**).”

Reviewer #3 (Remarks on code availability):

All the codes were derived from previous published studies. The authors have clarified that the novelty is not their deep learning methods. Thus, I did not review the codes.

Reviewers' Comments:

Reviewer #3:

Remarks to the Author:

The authors addressed my concerns.